# SHAP zero Explains Biological Sequence Models with Near-zero Marginal Cost for Future Queries

**Darin Tsui**
Georgia Institute of Technology
darint@gatech.edu

**Aryan Musharaf**
Georgia Institute of Technology
amusharaf3@gatech.edu

**Yigit Efe Erginbas**
UC Berkeley
erginbas@berkeley.edu

**Justin Singh Kang**
UC Berkeley
justin kang@berkeley.edu

**Amirali Aghazadeh**
Georgia Institute of Technology
amiralia@gatech.edu

## Abstract

The growing adoption of machine learning models for biological sequences has intensified the need for interpretable predictions, with Shapley values emerging as a theoretically grounded standard for model explanation. While effective for local explanations of individual input sequences, scaling Shapley-based interpretability to extract global biological insights requires evaluating thousands of sequences—incurring exponential computational cost per query. We introduce SHAP zero, a novel algorithm that amortizes the cost of Shapley value computation across large-scale biological datasets. After a one-time model sketching step, SHAP zero enables near-zero marginal cost for future queries by uncovering an underexplored connection between Shapley values, high-order feature interactions, and the sparse Fourier transform of the model. Applied to models of guide RNA efficacy, DNA repair outcomes, and protein fitness, SHAP zero explains predictions orders of magnitude faster than existing methods, recovering rich combinatorial interactions previously inaccessible at scale. This work opens the door to principled, efficient, and scalable interpretability for black-box sequence models in biology.

## 1 Introduction

The remarkable success of machine learning in modeling biological sequences, from DNA to proteins, has created an urgent need for interpretability tools that can reveal what these models have learned. Black-box sequence models now guide genome editing [1–3], protein design [4–6], and regulatory variant prediction [7–9], yet explaining their predictions remains prohibitively expensive at scale [10].

In this work, we study the problem of explaining black-box models that map a length-$n$ biological sequence to a real-valued prediction. We denote such models as $f : \mathbb{Z}_q^n \to \mathbb{R}$, where each sequence $\mathbf{x} \in \mathbb{Z}_q^n$ consists of symbols from an alphabet of size $q$ (e.g., 4 for DNA, 20 for proteins). Our goal is to explain the predictions of $f$ over $Q$ many input sequences $\mathbf{x}_1, \mathbf{x}_2, \ldots, \mathbf{x}_Q$ at scale. A popular interpretability framework is SHapley Additive exPlanations (SHAP) [11], which assigns an additive importance score $I_{\mathbf{x}_i}^{SV}(j)$ to each input feature $j$ in a given sequence $\mathbf{x}_i$. These scores are grounded in cooperative game theory [12] and quantify the marginal contribution of feature $j$ across all subsets of other features: $I_{\mathbf{x}_i}^{SV}(j) = \sum_{T \subseteq D \setminus \{j\}} \frac{|T|!\,(|D|-|T|-1)!}{|D|!} \left[ v_{T \cup \{j\}}(\mathbf{x}_i) - v_T(\mathbf{x}_i) \right]$, where $D = \{1, 2, \ldots, n\}$ is the full feature set. The value function $v_T(\mathbf{x}_i)$ denotes the expected output when only features in $T$ are fixed and the rest are marginalized over: $v_T(\mathbf{x}_i) = \frac{1}{q^{|\bar{T}|}} \sum_{\mathbf{m}:\mathbf{m}_{\bar{T}} \in \mathbb{Z}_q^{|\bar{T}|}, \mathbf{m}_T = \mathbf{x}_{iT}} f(\mathbf{m})$. See Appendix A for more details on Shapley values.

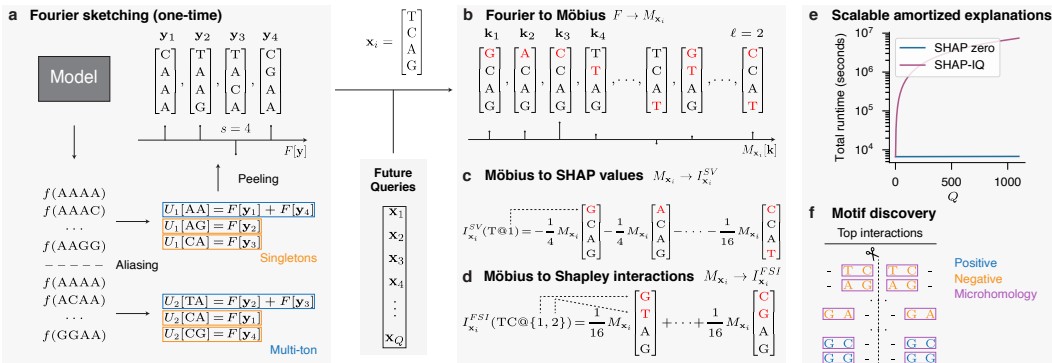

Figure 1: **Overview of SHAP zero. a**, SHAP zero pays a one-time cost to create a global Fourier sketch of $f$. This illustration shows $s = 4$ Fourier coefficients strategically aliased into multiple subsampled transforms $(U_1, U_2)$, and recovered by identifying singleton bins. **b**, For each future query $\mathbf{x}_1, \dots, \mathbf{x}_Q$, SHAP zero localizes the global sketch via the Möbius transform of order $\ell$, capturing query-specific feature interactions. This maps to **c**, SHAP values, and **d**, Shapley interactions. By marginalizing the cost of future queries, SHAP zero enables **e**, scalable amortized explanations and **f**, discovery of biological motifs at unprecedented scale.

Unfortunately, computing exact SHAP values requires evaluating $f$ on an exponential number of perturbed sequences, making it intractable even for modest $n$. Stochastic estimators [11, 13–18], such as KernelSHAP [11], reduce the cost with random sampling, while model-based approximators [11, 19–24] exploit internal structure but are often restricted to white-box access. Recent algorithms for Shapley interaction indices [25–29], such as SHAP-IQ [25], further increase this burden, requiring even more model evaluations.

These challenges are particularly severe in interpretation of biological sequence models, where one often seeks explanations across thousands of query sequences in a dataset to gain a global insight about the underlying biological phenomenon. For instance, estimating SHAP values for about a thousand guide RNA sequences in the gene editing model TIGER took one day on our single NVIDIA RTX A6000 machine. Estimating third-order Faithful Shapley (Faith-Shap) interactions through SHAP-IQ is projected to take over 80 days—highlighting a fundamental scalability bottleneck.

This raises a **key question**: can we amortize the cost of explaining sequence models across queries?

**A Fourier view of SHAP**. We observe that model evaluations used to compute Shapley explanations for one sequence contain information that can be reused to explain other sequences. This motivates a different paradigm: rather than estimating SHAP values and interactions de novo for each sequence, we can first sketch the black-box model globally, then compute local explanations from this sketch.

To realize this, we uncover a powerful and underexplored connection between Shapley explanations and the model's Fourier transform through the Möbius transform. The Fourier basis provides a global, query-agnostic decomposition of $f$, and if $f$ is compressible in this basis—as many biological sequence models are [30–32]—we can estimate its top-$s$ coefficients efficiently. To obtain localized feature contributions, we introduce a novel mapping from the Fourier transform to the localized Möbius transform around each query, reorganizing the global sketch to isolate interacting features. This global-to-local framework enables efficient amortized computation of Shapley explanations across future queries without recomputing model evaluations, and, as a result, allows for the discovery of high-order features (motifs) at unprecedented scales (Fig. 1).

**Contributions.** We present SHAP zero, a new algorithm for efficient model explanation of black-box sequence models via sparse Fourier sketching. After an initial one-time sketching phase, SHAP zero estimates SHAP values and Shapley interactions with *near-zero additional cost per query sequence*.

Our key contributions are:

- **Fourier sketching for biological sequences.** We develop a sample-efficient algorithm to recover the top-$s$ Fourier coefficients of $f$, defined over $\mathbb{Z}_q^n$, with sample complexity $\mathcal{O}(sn^2)$ and runtime $\mathcal{O}(sn^3)$, that is, only a polynomial dependence on the input sequence length $n$.

- **Formal connection between Fourier and SHAP through the Möbius transform.** We introduce a formal pipeline for mapping global Fourier coefficients to the localized Möbius transform, and then to SHAP values and interactions, enabling principled amortized explanations.

- **Scalable amortized explanation and motif discovery.** We conduct large-scale experiments to explain two genomic models, TIGER [33] and inDelphi [34], and the protein language model Tranception [35] with SHAP zero. We demonstrate that SHAP zero estimates feature interactions with an amortized computational cost *up to 1000 times faster* than current methods. SHAP zero reveals GC content of the seed region in TIGER, microhomologous motifs in inDelphi, and epistatic interactions in Tranception as predictive features, a task previously inaccessible due to the combinatorial space of possible feature interactions. Software for SHAP zero is available at https://github.com/amirgroup-codes/shap-zero.

## 2 Background

**Set Möbius Transform.** We begin by first introducing the classic Möbius transform in relationship to SHAP values and interactions. The classic Möbius transform formulation marginalizes the contributions between interactions of sets [36, 37]. However, our global sketch via the Fourier transform captures interactions between *features* in a sequence. For this reason, we will first introduce the classic Möbius transform formulation, which we will refer to as the set Möbius transform for clarity, and introduce the SHAP and Faith-Shap interaction equation in terms of the set Möbius transform. Then, in a later section, we will introduce an extension of the Möbius transform for $q$-ary functions. The definition of the set Möbius transform is as follows.

**Definition 2.1.** (Set Möbius transform). Given a value function $v : 2^n \to \mathbb{R}$ and a set $S$, the set Möbius transform $a_{\mathbf{x}_i}(v, S)$ is defined using the forward and inverse transform:

$$\text{Forward:} \quad a_{\mathbf{x}_i}(v, S) = \sum_{T \subseteq S} (-1)^{|S|-|T|} v_T(\mathbf{x}_i), \qquad \text{Inverse:} \quad v_T(\mathbf{x}_i) = \sum_{S \subseteq T} a_{\mathbf{x}_i}(v, S). \quad (1)$$

Each set Möbius transform coefficient $a_{\mathbf{x}_i}(v, S)$, where $S \subseteq T$, represents the marginal contribution of the subset $S$, given a value function $v$, in determining the score $v_T(\mathbf{x}_i)$. Reconstructing the original score $v_T(\mathbf{x}_i)$ from all subsets $S \subseteq T$ requires taking a linear sum over all the marginal contributions.

The set Möbius transform naturally provides a bridge between SHAP values [38] and Faith-Shap interactions [26]. We give both equations below, where, given a maximum interaction order $\ell$, $I_{\mathbf{x}_i}^{FSI}(T)$ is the Faith-Shap interaction index over a set $T$ in a given sequence $\mathbf{x}_i$:

$$I_{\mathbf{x}_i}^{\text{SV}}(j) = \sum_{T \subseteq D \setminus \{j\}} \frac{1}{|T \cup \{j\}|} a_{\mathbf{x}_i}(v, T \cup \{j\}), \qquad I_{\mathbf{x}_i}^{\text{FSI}}(T) = a_{\mathbf{x}_i}(v, T) \quad (2)$$

**Value Function in $q$-ary Functions.** The computation of SHAP values and Faith-Shap interactions depends on the value function $v_T(\mathbf{x}_i)$, which quantifies how much the features in $T$ contribute toward the prediction of $f(\mathbf{x}_i)$. In this paper, we define the value function as the expectation of $f(\mathbf{x}_i)$ with respect to $\mathbf{x}_{i\bar{T}}$: $v_T(\mathbf{x}_i) = \mathbb{E}_{p(\mathbf{x}_{i\bar{T}})}[f(\mathbf{x}_i)]$. Here, we take the expectation over the marginal distribution $p(\mathbf{x}_{i\bar{T}})$ for the missing inputs. $\mathbb{E}_{p(\mathbf{x}_{i\bar{T}})}$ can be computed for any Shapley problem by approximating the missing inputs $\mathbf{x}_{i\bar{T}}$. However, in $q$-ary functions, we can compute this equation exactly because we are constrained to $q$ alphabets at each site; every possible missing input must be in $\mathbb{Z}_q^{\bar{T}}$. By assuming that every possibility of $\mathbf{x}_{i\bar{T}} \in \mathbb{Z}_q^{\bar{T}}$ is equally likely, the value function reduces to taking an average contribution over $q^{|\bar{T}|}$ possible inputs.

## 3 The SHAP zero Algorithm

In this section, we detail the SHAP zero algorithm. SHAP zero estimates SHAP values and interactions in three steps: *(i)* Estimating Fourier coefficients, *(ii)* Computing the Möbius transform, and *(iii)* Finding SHAP values and Faith-Shap interactions. An overview of SHAP zero is provided in Algorithms 1 and 2 in Appendix B.

## 3.1 Estimating Fourier Coefficients

The first key step in SHAP zero is to compute the sparse Fourier transform of $f$ (Fig. 1a). Any function $f : \mathbb{Z}_q^n \to \mathbb{R}$ can be expressed in terms of its Fourier transform $F[\mathbf{y}]$ as:

$$f(\mathbf{m}) = \sum_{\mathbf{y} \in \mathbb{Z}_q^n} F[\mathbf{y}] \omega^{\langle \mathbf{m}, \mathbf{y} \rangle}, \quad \mathbf{m} \in \mathbb{Z}_q^n, \tag{3}$$

where $\omega = e^{\frac{2\pi j}{q}}$, and $\mathbf{y}$ is the frequency vector. The Fourier transform provides a global sketch of $f$, irrespective of the input query sequence.

Computing the Fourier transform exactly requires obtaining $q^n$ samples from $f$, which can be prohibitive for large values of $q$ or $n$. Fortunately, in practice, sequence models tend to have a sparse Fourier transform [30–32, 39, 40], meaning $F[\mathbf{y}]$ only has a few non-zero coefficients.

SHAP zero takes advantage of sparsity and pays a *one-time cost* to estimate the top-$s$ Fourier coefficients of $f$. We leverage structured subsampling in $f$ using patterns from sparse-graph codes [40–42], which implicitly hash Fourier coefficients into buckets. To do this in sequence models, SHAP zero creates $C$ many subsampling groups. For each subsampling group $c = 1 \ldots C$, we define a random subsampling matrix $\mathbf{M}_c \in \mathbb{Z}_q^{n \times b}$, where $b$ is the subsampling dimension. Additionally, we define a total of $P$ offsets $\mathbf{d}_{c,p} \in \mathbb{Z}_q^n$, where $p \in [P]$. The subsampled transform $U_{c,p}[\mathbf{j}]$, indexed by $\mathbf{j} \in \mathbb{Z}_q^b$, is computed as: $U_{c,p}[\mathbf{j}] = \frac{1}{B} \sum_{\boldsymbol{\ell} \in \mathbb{Z}_q^b} f(\mathbf{M}_c \boldsymbol{\ell} + \mathbf{d}_{c,p}) \omega^{\langle \mathbf{j}, \boldsymbol{\ell} \rangle}$.

By subsampling $f$, we leverage a classical result in signal processing, which states that regularly subsampling a signal in time introduces predictable *aliasing* patterns in the Fourier domain. Using $\mathbf{M}_c$ and $\mathbf{d}_{c,p}$, the induced aliasing structure satisfies:

$$U_{c,p}[\mathbf{j}] = \sum_{\mathbf{k}:\, \mathbf{M}_c^T \mathbf{k} = \mathbf{j}} F[\mathbf{k}] \, \omega^{\langle \mathbf{d}_{c,p}, \mathbf{k} \rangle}. \tag{4}$$

Thus, each $U_{c,p}[\mathbf{j}]$ is a linear combination of $F[\mathbf{k}]$ that are hashed into buckets defined by $\mathbf{M}_c$. Our goal is to maximize the number of $U_{c,p}[\mathbf{j}]$ that contain exactly one non-zero Fourier coefficient, which we call a *singleton*. Singletons allow direct recovery of the corresponding $F[\mathbf{k}]$. However, inevitably, some $U_{c,p}[\mathbf{j}]$ will contain more than one Fourier coefficient, which we call a *multi-ton*. In Fig. 1a, we alias the $s = 4$ non-zero Fourier coefficients into two buckets, $U_1$ and $U_2$ (where we drop the notation $p$ in $U_{c,p}$ for simplicity), assuming $\mathbf{d}_{c,p}$ is a vector of all zeros. $U_1$ generated two singletons containing $F[\mathbf{y}_1]$ and $F[\mathbf{y}_4]$ and one multi-ton, and $U_2$ similarly generated two singletons containing $F[\mathbf{y}_2]$ and $F[\mathbf{y}_3]$ and one multi-ton, allowing us to recover the entire Fourier transform.

In practice, though, the Fourier transform may not be able to be fully recovered in one shot. Therefore, we can resolve multi-tons to generate more singletons by modeling the aliasing structure across subsampling groups as a sparse bipartite graph. SHAP zero then maximizes the number of singletons recovered by applying a peeling procedure [40–42]. Upon recovery of a singleton, its contribution can be subtracted from adjacent multi-tons, potentially creating new singletons. This peeling process continues until no further singletons can be found. SHAP zero recovers the top-$s$ Fourier coefficients with a sample complexity of $\mathcal{O}(sn^2)$ and a computational complexity of $\mathcal{O}(sn^3)$. Appendix C contains more details on computing the sparse Fourier transform in sequence models.

## 3.2 Computing the Möbius Transform

The second step of SHAP zero is to map the sparse Fourier coefficients of $f$ to the Möbius transform of $f$ around an input query sequence $\mathbf{x}_i$ (Fig. 1b). We build on the binary Möbius transform formulation found in [43] for functions characterized in $f : \mathbb{Z}_2^n \to \mathbb{R}$, which we dub as the *q-ary Möbius transform*.

We first define the $q$-ary Möbius transform as a function of $f$. Then, we describe how it can be estimated via the Fourier transform. We denote $\mathbf{m} \leq \mathbf{k}$ to be $m_i = k_i$ or $m_i = 0$ for all of $i$, and subtraction of $\mathbf{k} - \mathbf{m}$ for $\mathbf{m} \leq \mathbf{k}$ by standard real field subtraction. The $q$-ary Möbius transform around the input sequence $\mathbf{x}_i$, $M_{\mathbf{x}_i}[\mathbf{k}]$, where $\mathbf{k}$ is the feature interaction vector, is defined as:

**Definition 3.1.** (*q-ary Möbius transform*). Given $f : \mathbb{Z}_q^n \to \mathbb{R}$, the $q$-ary Möbius transform $M_{\mathbf{x}_i}[\mathbf{k}]$, where $\mathbf{k}$ is the feature interaction vector, can be defined as,

$$M_{\mathbf{x}_i}[\mathbf{k}] = \sum_{\mathbf{m} \leq \mathbf{k}} (-1)^{\|\mathbf{k} - \mathbf{m}\|_0} f((\mathbf{m} + \mathbf{x}_i) \bmod q), \quad \mathbf{m}, \mathbf{k} \in \mathbb{Z}_q^n, \tag{5}$$

with its inverse defined as,

$$f((\mathbf{m} + \mathbf{x}_i) \bmod q) = \sum_{\mathbf{k} \leq \mathbf{m}} M_{\mathbf{x}_i}[\mathbf{k}]. \tag{6}$$

$M_{\mathbf{x}_i}[\mathbf{k}]$ captures how the output of $f$ changes when features in $\mathbf{x}_i$ are perturbed (Fig. 1b). The modulo operation is used to ensure the inequalities $\mathbf{m} \leq \mathbf{k}$ and $\mathbf{k} \leq \mathbf{m}$ hold when $\mathbf{x}_i$ is encoded as a vector in $\mathbb{Z}_q^n$. Appendix D illustrates the differences in computing the set and $q$-ary Möbius transform.

Without assumptions, the $q$-ary Möbius transform scales with $\mathcal{O}(q^n)$ computations, which is intractable for large values of $q$ or $n$. In order to bypass these computational issues, we map the top-$s$ Fourier coefficients to $M_{\mathbf{x}_i}[\mathbf{k}]$. In Appendix E.1, we prove the following Proposition:

**Proposition 3.2.** *(Fourier transform to $q$-ary Möbius transform). Given the top-s Fourier coefficients $F[\mathbf{y}]$ with a maximum order of $\ell$ and the input query sequence $\mathbf{x}_i$, $M_{\mathbf{x}_i}[\mathbf{k}]$ is defined as:*

$$M_{\mathbf{x}_i}[\mathbf{k}] = \sum_{\mathbf{y} \in \mathbb{Z}_q^n} F[\mathbf{y}] \omega^{\langle \mathbf{x}_i, \mathbf{y} \rangle} \left( \sum_{\mathbf{m} \leq \mathbf{k}} (-1)^{\|\mathbf{k} - \mathbf{m}\|_0} \omega^{\langle \mathbf{m}, \mathbf{y} \rangle} \right), \quad \forall \mathbf{k} \in \mathbb{Z}_q^n, \tag{7}$$

*where the computational complexity of Equation (7) scales with $\mathcal{O}(s^2 (2q)^\ell)$.*

Here, $\ell$ is the maximum order of Fourier interactions recovered by SHAP zero and, in practice, bounded by $\ell \leq 5$ [32]. Since $F[\mathbf{y}]$ provides a global sketch of the model irrespective of $\mathbf{x}_i$, the shifting property of the Fourier transform (see Equation (21) in Appendix E.1) is first used to align $F[\mathbf{y}]$ with $\mathbf{x}_i$. Then, Equation (3) is plugged into Equation (5) and simplified to obtain the final result.

### 3.3 Finding SHAP Values and Faith-Shap Interactions

The last step of SHAP zero is to map $M_{\mathbf{x}_i}[\mathbf{k}]$ to SHAP values and Faith-Shap interactions. We have a direct relationship going from the set Möbius transform to SHAP values and Faith-Shap interactions. Therefore, computing Shapley explanations for $q$-ary functions first requires a conversion from the $q$-ary Möbius to the set Möbius transform. In Appendix E.2, we prove the following Proposition:

**Proposition 3.3.** *($q$-ary Möbius transform to set Möbius transform). Computing the set Möbius transform $a_{\mathbf{x}_i}(v, S)$ given $M_{\mathbf{x}_i}[\mathbf{k}]$ is defined as:*

$$a_{\mathbf{x}_i}(v, S) = (-1)^{|S|} \sum_{\mathbf{k}: \mathbf{k}_S > \mathbf{0}_{|S|}} \frac{1}{q^{\|\mathbf{k}\|_0}} M_{\mathbf{x}_i}[\mathbf{k}]. \tag{8}$$

The proof follows two major steps. After computing $M_{\mathbf{x}_i}[\mathbf{k}]$, the inverse $q$-ary Möbius transform from Equation (6) is plugged into the $q$-ary value function. Then, the $q$-ary function is plugged into the forward set Möbius transform to obtain Equation (8).

Using Proposition 3.3 and the $I_{\mathbf{x}_i}^{SV}(j)$ equation from Equation (2), we prove the following Theorem:

**Theorem 3.4.** *($q$-ary Möbius transform to SHAP values). Given $f : \mathbb{Z}_q^n \to \mathbb{R}$ and $M_{\mathbf{x}_i}[\mathbf{k}]$, the SHAP value equation can be written as:*

$$I_{\mathbf{x}_i}^{SV}(j) = - \sum_{\mathbf{k}: k_j > 0} \frac{1}{\|\mathbf{k}\|_0 q^{\|\mathbf{k}\|_0}} M_{\mathbf{x}_i}[\mathbf{k}], \tag{9}$$

*with a computational complexity of $\mathcal{O}(q^\ell)$, where $\ell$ is the maximum $q$-ary Möbius coefficients order.*

The proof can be found in Appendix E.3. Here, the $>$ operation is defined over the standard real field in $\mathbb{Z}_q^n$ and the $L_0$ norm $\|\mathbf{k}\|_0$ counts the number of nonzero elements in $\mathbf{k}$ (Fig. 1c). $I_{\mathbf{x}_i}^{SV}(j)$ can be interpreted as the contribution of the $j^{\text{th}}$ feature to the prediction of $f(\mathbf{x}_i)$; when $I_{\mathbf{x}_i}^{SV}(j) > 0$, substituting the $j^{\text{th}}$ feature to a different feature will, on average, decrease the value of $f(\mathbf{x}_i)$, and vice versa. Therefore, $I_{\mathbf{x}_i}^{SV}(j)$ sums over $q$-ary Möbius transform coefficients, which capture the marginal effects of all subsets of features on $f(\mathbf{x}_i)$, with a negative sign.

Similarly, SHAP zero can map the $q$-ary Möbius transform to $I_{\mathbf{x}_i}^{FSI}(T)$ (Fig. 1d). Since the $I_{\mathbf{x}_i}^{FSI}(T)$ equation in Equation (2) is already written in terms of $a_{\mathbf{x}_i}(v, S)$:

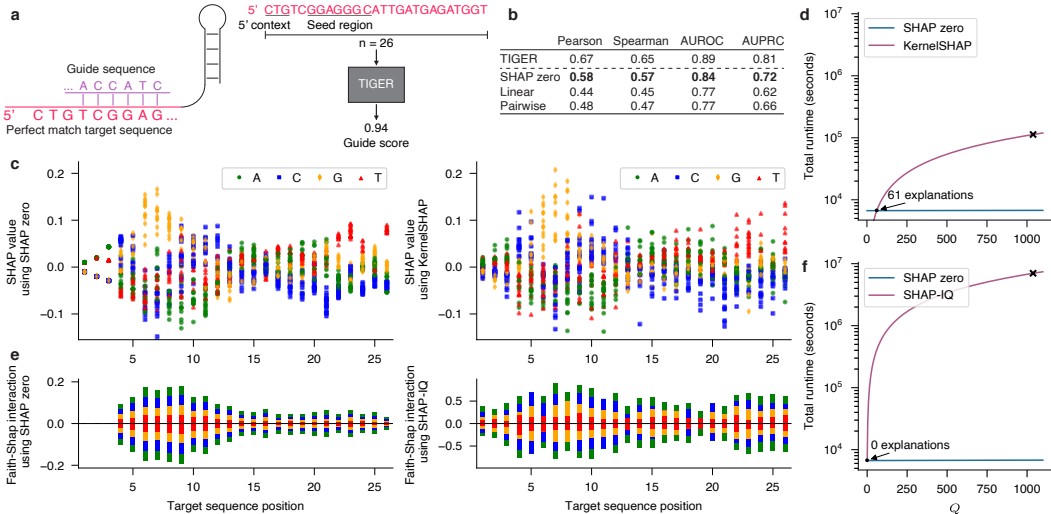

Figure 2: **SHAP zero enables scalable amortized explanations in TIGER. a**, TIGER [33] predicts the guide score of $n = 26$ length target sequences. **b**, The estimated top Fourier coefficients by SHAP zero outperform linear and pairwise models in predicting the guide scores in a held-out set. **c**, SHAP value estimates reveal high agreements ($\rho = 0.83$) between SHAP zero and KernelSHAP. **d**, Total runtime of SHAP zero against KernelSHAP is marked by $\times$ in plots that depict the computational cost versus the number of explained sequences in both algorithms. **e**, Histogram of Faith-Shap interactions from SHAP zero compared to SHAP-IQ (see Appendix F). **f**, Total runtime of SHAP zero versus SHAP-IQ in TIGER demonstrate that SHAP zero is more than 1000-fold faster.

**Theorem 3.5.** *(q-ary Möbius transform to Faith-Shap). Given $f : \mathbb{Z}_q^n \to \mathbb{R}$ and $M_{\mathbf{x}_i}[\mathbf{k}]$ with a maximum order $\ell$, the $\ell^{th}$ order Faith-Shap interaction index equation can be written as:*

$$I_{\mathbf{x}_i}^{FSI}(T) = (-1)^{|T|} \sum_{\mathbf{k}:\mathbf{k}_T > \mathbf{0}_{|T|}} \frac{1}{q^{\|\mathbf{k}\|_0}} M_{\mathbf{x}_i}[\mathbf{k}], \tag{10}$$

*with a computational complexity of $\mathcal{O}(q^\ell)$.*

The proof in Appendix E.4 hinges on the fact that the Faith-Shap interaction index and the $q$-ary Möbius transform are both $\ell^{th}$ order. If this is the case, computing Faith-Shap is equivalent to converting from the $q$-ary Möbius transform to the set Möbius transform from Equation (8). Faith-Shap can be seen as a generalization of the SHAP value for feature interactions [26]; when $\ell = 1$, computing Faith-Shap interactions becomes identical to computing SHAP values using Equation (9).

### 3.4 Computational Complexity of SHAP zero

After paying a one-time cost to find the top-$s$ Fourier coefficients, which scale with $poly(n)$, SHAP zero explanations do not scale with $n$. By bounding the computational complexity of Proposition 3.2, Theorem 3.4, and Theorem 3.5 by $\ell$, we remove the computational dependency on $n$. The dependence on $\ell$ does not impact SHAP zero in practice, since $\ell$ is typically a small constant for sequence models. Additionally, while Equations (7), (9), and (10) depend on $q$, in biological sequences, we are restricted to either $q = 4$ nucleotides or $q = 20$ amino acids. These combined enable SHAP zero to explain a new query sequence with near-zero marginal cost (essentially free).

## 4 Experiments

**Explaining Guide RNA Binding.** TIGER [33] is a convolutional neural network trained to predict the binding efficiency of CRISPR-Cas13d guide RNA (gRNA) to a target DNA sequences (Fig. 2a). We considered length-$n = 26$ input sequences to the model that perfectly match the target and guide RNA sequences that reverse complement the target. Positions 1-3 correspond to the additional 5' context given to the target sequence and positions 6-12 of the target sequence correspond to the

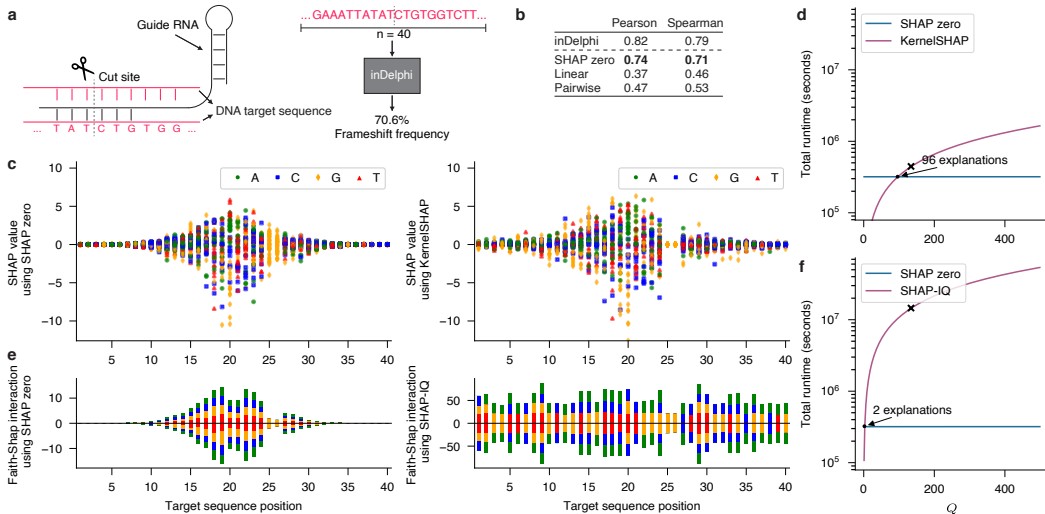

Figure 3: **SHAP zero reveals high-order motifs in inDelphi. a**, inDelphi [34] predicts DNA repair outcomes in $n = 40$ length sequences. **b**, Recovered Fourier coefficients outperform linear and pairwise models in repair outcomes in a held-out set. AUROC and AUPRC are not reported due to the regression nature of the model (see Appendix G.7). **c**, SHAP zero and KernelSHAP estimates reveal the importance of nucleotides around the cut site. **d**, Total runtime of SHAP zero versus KernelSHAP. **e**, Histogram of Faith-Shap interactions in SHAP zero compared to SHAP-IQ (see Appendix F). High-order feature interactions identified by SHAP zero reveal the importance of microhomology patterns around the cut site (see Appendix F). **f**, Total runtime of SHAP zero versus SHAP-IQ.

nucleotides that bind with the seed region in the gRNA: a short segment that facilitates the initial pairing with the target [44] and is especially sensitive to mutations [45].

To assess the accuracy of the recovered Fourier coefficients from SHAP zero, we predict the experimental guide scores of 1038 held-out target sequences (Fig. 2b). SHAP zero's recovered Fourier coefficients predict experimental guide scores with a Pearson correlation of $\rho = 0.58$, outperforming the linear ($\rho = 0.44$) and pairwise ($\rho = 0.48$) model with L2 regularization, suggesting that it successfully captures high-order ($\ell > 1$) feature interactions. SHAP zero and KernelSHAP estimates are in high agreements ($\rho = 0.83$), with 47% and 56% of their top positive estimates attributed to C and G nucleotides in the seed region, respectively, highlighting the GC content as a predictive feature (see Fig. 2c and Appendix F). However, it took us an amortized time of only 6.5 seconds to compute SHAP zero values compared to 109 seconds to find KernelSHAP values (Fig. 2d), a 17-fold speedup.

We then explained the high-order feature interactions with SHAP zero and compared it with SHAP-IQ. The overlapping interactions in SHAP zero and SHAP-IQ, constituting 95% and 5% of the total interactions, respectively, correlate with $\rho = 0.79$. We expected most interactions to be located within the seed region [33, 46]; however, we observed that only 34% of SHAP-IQ's interactions are within the seed region, compared to 52% of SHAP zero's interactions (Fig. 2e). Besides, estimating the Faith-Shap interactions took an amortized time of 6.5 seconds for SHAP zero compared to 2 hours for SHAP-IQ (Fig. 2f), suggesting that SHAP zero is more than 1000-fold faster and enables a biologically feasible explanation of feature interactions.

**Explaining DNA Repair Outcome.** inDelphi [34] is a machine learning model trained to predict DNA repair outcomes after CRISPR-Cas9-induced double-stranded breaks, including the probability of Cas9 inducing a one- or two-base pair insertion or deletion (frameshift frequency) (Fig. 3a). We considered length-$n = 40$ input DNA target sequences, where the cut site occurs between positions 20 and 21, a region that is considered to have the most predictive power [30, 34, 47].

We used the recovered Fourier coefficients from SHAP zero to predict the experimental frameshift frequencies of 84 held-out target sequences (Fig. 3b). SHAP zero ($\rho = 0.74$) significantly outperformed linear ($\rho = 0.37$) and pairwise models ($\rho = 0.47$), suggesting that inDelphi's interactions cannot be explained with just 2nd order interactions. SHAP zero and KernelSHAP estimates are in high agreement again ($\rho = 0.84$) (see Fig. 3c and Appendix F). This time, it took SHAP zero an

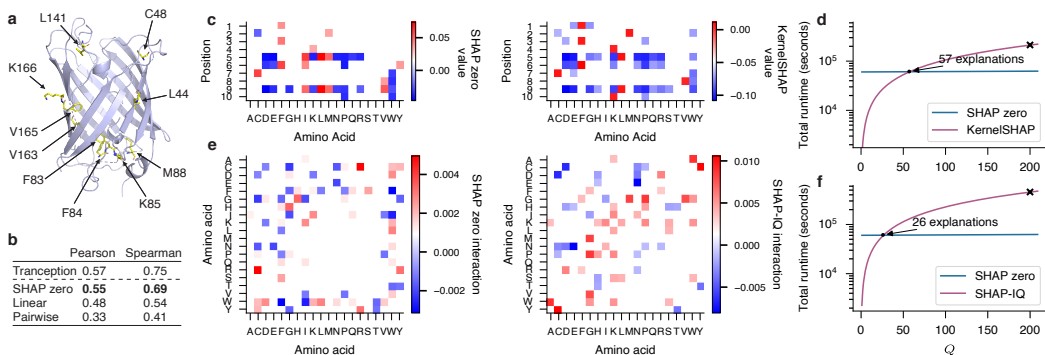

Figure 4: **SHAP zero uncovers epistatic interactions in Tranception. a**, We analyze the green florescence protein over $n = 10$ epistatic sites from [49]. **b**, SHAP zero Fourier coefficients outperform linear and pairwise models in predicting protein function in a held-out set. **c**, SHAP zero and KernelSHAP estimates ($\rho = 0.97$) reveal the importance of secondary structure promoters. **d**, Total runtime of SHAP zero versus KernelSHAP. **e**, Heatmap of Faith-Shap interactions run over 200 sequences with SHAP zero and 50 sequences with SHAP-IQ. **f**, Total runtime of SHAP zero versus SHAP-IQ.

amortized time of 40 minutes to estimate the SHAP values compared to KernelSHAP's 55 minutes (Fig. 3d). The difference in amortized time is smaller here than in TIGER due to the smaller set of query sequences.

We then used SHAP zero and SHAP-IQ to extract high-order feature interactions. The overlapping interactions (constituting 85% and 3% of the total interactions in SHAP zero and SHAP-IQ, respectively) correlate with $\rho = 0.74$. However, just 17% of SHAP-IQ's interactions are within three nucleotides of the cut site compared to 55% of SHAP zero's interactions (Fig. 3e). Similar to TIGER, we noticed a significant speedup in amortized time when using SHAP zero compared to SHAP-IQ: SHAP zero took 40 minutes compared to 37 hours from SHAP-IQ (Fig. 3f).

We performed a detailed analysis of the top SHAP zero feature interactions of inDelphi (see Appendix F). Our analysis reveals repeating motifs around the cut site as key high-order features predictive of DNA repair [30, 34]. These high-order features can be attributed to microhomology patterns that mediate long deletions around the cut site (Fig. 1f) [48]. Notably, a top feature identified by SHAP zero enriching the frameshift frequency is the repetition of the GC motif at sites 17-18 and 21-22. This feature captures the microhomology-mediated deletion of four nucleotides at 17-20, which results in a one-base pair frameshift. Another top feature identified by SHAP zero depleting the frameshift frequency is the repetition of the TC motif at sites 18-19 and 21-22. This feature captures the microhomology-mediated deletion of three nucleotides at 18-20, which results in no frameshift. These examples highlight the unique capability of SHAP zero in extracting biologically meaningfully high-order interactions, which was previously inaccessible due to the space of feature interactions.

**Capturing Epistatic Interactions in Protein Language Models.** Protein language models have been shown to implicitly encode high-order (otherwise known as epistatic) interactions between amino acid sites [32, 35, 50]. This motivates the question: can we explicitly extract these interactions using SHAP zero? To explore this, we analyze Tranception [35], a protein language model trained to predict fitness from an amino acid sequence. Specifically, we focus on the green fluorescent protein (GFP), which has been shown to exhibit high-order interactions related its function [49, 51]. We focus on evaluating predictions from Tranception over $n = 10$ epistatic positions from [49] (Fig. 4a).

Similar to the genomics experiments, SHAP zero's recovered Fourier coefficients ($\rho = 0.55$) outperformed linear ($\rho = 0.48$) and pairwise ($\rho = 0.33$) models in predicting protein function, further reinforcing its ability to capture interactions (Fig. 4b). SHAP zero and KernelSHAP also showed strong agreement in their estimates ($\rho = 0.97$) (Fig. 4c). Despite $q^n$ being on the order of $\sim 10^{13}$, SHAP zero was still 3× faster in amortized time (5.2 vs. 17.7 minutes, respectively) (Fig. 4d). Comparing SHAP zero to SHAP-IQ, the overlapping interactions correlated with $\rho = 0.61$ (Fig. 4e), while running 7× faster in amortized time (5.2 vs. 38.1 minutes, respectively) (Fig. 4f).

We analyzed SHAP zero's values (see Appendix F) and found that amino acids known to promote $\beta$ sheet stability, such as isoleucine (I) and valine (V), and alpha helix stability, such as leucine (L) and methionine (M), were strongly associated with increased fitness [52]. Notably, proline (P) stood out as a dominant amino acid associated with reduced fitness, aligning with its well-established role as a secondary structure breaker due to its rigid backbone [53, 54]. However, SHAP zero's interaction values revealed a more complex picture: despite its overall negative effect, proline can participate in *positive* epistatic interactions (e.g. when position 5 interacts with positions 7 and 9) when paired with specific residues like lysine (K) or asparagine (N), suggesting its disruptive influence can be contextually mitigated. Lysine also displayed strong context dependence, appearing in both favorable and unfavorable interactions. These results highlight SHAP zero's ability to uncover epistatic interactions at scale, a feat previously unattainable. Details about all experimental setups can be found in Appendix G[1].

## 5   Discussion

**Contrasting Different Amortization Strategies.** From the amortization perspective, SHAP zero is closely related to SHAP estimation algorithms that use surrogate models to speed up computations [55–57]. One such algorthim, FastSHAP [55], trains a surrogate model to predict model outputs given a partially masked input. Although FastSHAP is theoretically optimal when the surrogate achieves the global minimizer [58], our results show poor empirical correlation with KernelSHAP on TIGER ($\rho = -0.39$), inDelphi ($\rho = 0.08$), and Tranception ($\rho = 0.03$) (Appendix F). We attribute this failure to the nature of sequence models, where model outputs are highly sensitive to specific masked features (e.g., seed regions in TIGER, cut sites in inDelphi, and epistatic positions in Tranception). Training a reliable surrogate would require model evaluations over nearly all unmasked combinations, which is infeasible. This contrasts with data modalities such as images, where the model output is rarely a function of a few masked pixels. SHAP zero, on the other hand, solely relies on the assumption that the mode is compressible, leading to an accurate and less biased amortization strategy with comparable speedup. Other methods that offer amortization assume that the input is binary ($q = 2$), which are unsuitable for sequence models [59].

**Explainability Trade-offs in White-box Methods.** White-box explainability methods, such as DeepSHAP [11] and DeepLIFT [19], leverage the model architecture to estimate SHAP values extremely efficiently; however, they naturally fall short in *(i)* estimating high-order feature interactions, and *(ii)* black-box settings with only query access, which is the focus of this work. Regardless, SHAP zero due to its amortized nature, outperforms these methods in computational cost *asymptotically*. Extrapolating the amortized runtimes, our results in TIGER demonstrate that SHAP zero is more efficient when the queries exceed 25,000 sequences (see Appendix F).

**Limitations.** SHAP zero provides accurate and amortized Shapley estimates under the assumption that the Fourier transform is sparse. However, this does not hinder the generalizability of SHAP zero. In sequence models, sparsity is more than just a theoretical assumption; it emerges from models learning the biophysical properties underlying sequence-function relationships. Sparsity in Fourier transform reflects the presence of only a select few higher-order, epistatic interactions [60], a phenomenon consistent with biological studies [39, 51, 61]. Nevertheless, we emphasize that SHAP zero can *adapt* to varying levels of sparsity by tuning the number of samples while still amortizing explanation costs. For sparser models, SHAP zero requires fewer samples; for denser models, SHAP zero requires more samples. We can always take more samples to combat lack of sparsity. Our empirical results confirm this: SHAP zero took 4 million samples for inDelphi, 6 million samples for TIGER, and 800,000 samples for Tranception, but successfully amortized explanation costs across all settings. We provide detailed documentation on how to tune SHAP zero for future sequence models in Appendix G.1.

Additionally, SHAP zero's strategy for marginalizing absent features is closest to the uniform approach proposed in [62]. Estimating the top-$s$ Fourier coefficients by subsampling the sequence space is made under the key assumption that all $q$ alphabets are equally probable. This contrasts with other Shapley-based algorithms [11, 25, 26, 28, 62], which marginalize absent features based on the empirical training distribution. For instance, in Tranception, the empirical distribution of amino acids

---

[1]Code and implementation for conducting the experiments can be found at `https://github.com/amirgroup-codes/shapzero`. Refer to the Appendix for more details.

in real-world protein sequences used for KernelSHAP heavily favors a small subset of amino acids (less than 5), biasing KernelSHAP to produce more negative SHAP estimates when encountering rare amino acid substitutions (see Appendix F). While SHAP zero and KernelSHAP estimates remain well correlated, they approximate slightly different quantities due to their marginalization assumptions. An interesting extension would be to marginalize absent features using the training data distribution, which would necessitate the development of sparse Fourier algorithms capable of subsampling the sequence space while accounting for this distribution.

Despite these differences, SHAP zero provides sufficient explanations to identify both impactful amino acids and epistatic interactions. Comparing the top SHAP values from SHAP zero and KernelSHAP (Tables 8 and 9), SHAP zero recovers all of the top positive amino acids and 17 of the 20 top negative amino acids identified by KernelSHAP, with the three missed negatives appearing in the bottom five of KernelSHAP's ranking. Thus, although SHAP zero tends to assign negative contributions more conservatively, it still captures the most impactful amino acids. To further assess SHAP zero's reliability, we examined the top five positive and negative interactions from Table 10 and verified that these interactions are epistatic in Tranception (see Appendix F.1 for details). Taken together, these results demonstrate that SHAP zero reliably uncovers impactful amino acids and combinatorial interactions at scale.

## 6 Conclusion

In this paper, we introduced SHAP zero, an algorithm for estimating Shapley values and interactions with a near-zero additional cost for future queried sequences after paying an initial up-front cost to find the Fourier transform. Our theoretical analysis and large-scale experiments show that SHAP zero significantly reduces amortized computational cost by orders of magnitude while identifying nearly all predictive motifs. Overall, SHAP zero expands our toolkit in the explainability of biological sequence models and, more broadly, in problems with a combinatorial nature. Our work will further encourage interdisciplinary algorithms and theoretical frameworks at the intersection of signal processing, coding theory, and algebraic geometry for the benefit of explainability in machine learning.

## 7 Acknowledgment

This research was supported by the National Science Foundation (NSF) Graduate Research Fellowship Program (GRFP), the Parker H. Petit Institute for Bioengineering and Biosciences (IBB) interdisciplinary seed grant, the Institute of Matter and Systems (IMS) Exponential Electronics seed grant, and Georgia Institute of Technology start-up funds.

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

# A   Computing Shapley Values in $q$-ary Functions

The exact computation of Shapley values requires querying $f$ over an exponential number of query sequences. To illustrate this, we will walk through an example of computing the Shapley value $I_{\mathbf{x}_i}^{SV}(j=1)$ for a $q$-ary function that takes as an input of alphabet size $q=2$ and sequence length of $n=3$. For brevity, we will drop the notation $\mathbf{x}_i$.

For simplicity, let us assume the following function for $f$ and we are trying to explain the sequence $\mathbf{x} = [0,1,0]$:

$$f(\mathbf{m}) = m_1 + m_2 + m_3,$$

where $m_i$ indicates the $i^{\text{th}}$ coordinate of $\mathbf{m}$. In this example, it follows from $n=3$ that $D = \{1,2,3\}$. The total possible subsets $T \subseteq D\backslash\{j=1\}$ can then be written out as: $[\emptyset, \{2\}, \{3\}, \{2,3\}]$, where $\emptyset$ is the empty set. Expanding out the Shapley value equation from the Introduction, we then have the following:

$$
\begin{aligned}
I^{SV}(j=1) &= \frac{0!(3-0-1)!}{3!}[v_{\{1\}}(\mathbf{x}) - v_\emptyset(\mathbf{x})] \\
&+ \frac{1!(3-1-1)!}{3!}[v_{\{1,2\}}(\mathbf{x}) - v_{\{2\}}(\mathbf{x})] \\
&+ \frac{1!(3-1-1)!}{3!}[v_{\{1,3\}}(\mathbf{x}) - v_{\{3\}}(\mathbf{x})] \\
&+ \frac{2!(3-2-1)!}{3!}[v_{\{1,2,3\}}(\mathbf{x}) - v_{\{2,3\}}(\mathbf{x})] \\
&= \frac{1}{3}[v_{\{1\}}(\mathbf{x}) - v_\emptyset(\mathbf{x})] + \frac{1}{6}[v_{\{1,2\}}(\mathbf{x}) - v_{\{2\}}(\mathbf{x})] \\
&+ \frac{1}{6}[v_{\{1,3\}}(\mathbf{x}) - v_{\{3\}}(\mathbf{x})] + \frac{1}{3}[v_{\{1,2,3\}}(\mathbf{x}) - v_{\{2,3\}}(\mathbf{x})].
\end{aligned}
$$

From here, we compute each value function, which requires fixing features in $T$ and marginalizing the rest.

$$
\begin{aligned}
v_\emptyset(\mathbf{x}) &= \frac{1}{2^3}(f([0,0,0]) + f([1,0,0]) + f([0,1,0]) + f([0,0,1]) + f([1,1,0]) \\
&+ f([1,0,1]) + f([0,1,1]) + f([1,1,1])) = \frac{3}{2}. \\
v_{\{1\}}(\mathbf{x}) &= \frac{1}{2^2}(f([0,0,0]) + f([0,1,0]) + f([0,0,1]) + f([0,1,1])) = 1. \\
v_{\{2\}}(\mathbf{x}) &= \frac{1}{2^2}(f([0,1,0]) + f([1,1,0]) + f([0,1,1]) + f([1,1,1])) = 2. \\
v_{\{3\}}(\mathbf{x}) &= \frac{1}{2^2}(f([0,0,0]) + f([1,0,0]) + f([0,1,0]) + f([1,1,0])) = 1. \\
v_{\{1,2\}}(\mathbf{x}) &= \frac{1}{2}(f([0,1,0]) + f([0,1,1])) = \frac{3}{2}. \\
v_{\{1,3\}}(\mathbf{x}) &= \frac{1}{2}(f([0,0,0]) + f([0,1,0])) = \frac{1}{2}. \\
v_{\{2,3\}}(\mathbf{x}) &= \frac{1}{2}(f([0,1,0]) + f([1,1,0])) = \frac{3}{2}. \\
v_{\{1,2,3\}}(\mathbf{x}) &= \frac{1}{2^0}(f([0,1,0])) = 1.
\end{aligned}
$$

Due to the reliance of $q$ in the value function, we are required to loop over all $q^n$ sequences. Substituting all of these value functions into our Shapley value equation:

$$
\begin{aligned}
I^{SV}(j=1) &= \frac{1}{3}[1 - \frac{3}{2}] + \frac{1}{6}[\frac{3}{2} - 2] + \frac{1}{6}[\frac{1}{2} - 1] + \frac{1}{3}[1 - \frac{3}{2}] \\
&= -\frac{1}{6} - \frac{1}{12} - \frac{1}{12} - \frac{1}{6} = -\frac{1}{2}.
\end{aligned}
$$

# B  The SHAP zero Algorithm

---

**Algorithm 1** SHAP zero: Estimating Fourier Coefficients (One-time Cost)

---

**Require:** Alphabet size $q$, Subsampling dimension $b$, Number of subsampling groups $C$, Number of offsets $P$, Subsampling matrices $\{\mathbf{M}_c\}_{c\in[C]}$, Offsets $\{\mathbf{d}_{c,p}\}_{c\in[C],p\in[P]}$

 1: $F \leftarrow \emptyset$
 2: {Initialize recovered Fourier coefficients}
 3: **for** each $c \in [C]$ **do**
 4:     **for** $p \in [P]$ **do**
 5:         **for** each $\mathbf{j} \in \mathbb{Z}_q^b$ **do**
 6:             $U_{c,p}[\mathbf{j}] \leftarrow \frac{1}{q^b} \sum_{\boldsymbol{\ell} \in \mathbb{Z}_q^b} f(\mathbf{M}_c \boldsymbol{\ell} + \mathbf{d}_{c,p}) \, \omega^{\langle \mathbf{j}, \boldsymbol{\ell} \rangle}$
 7:         **end for**
 8:     **end for**
 9: **end for**
10: Build sets $\mathcal{S} = \{(c,\mathbf{j}) : U_{c,p}[\mathbf{j}] \text{ is a singleton}\}$, and $L = \{(c,\mathbf{j}) : U_{c,p}[\mathbf{j}] \text{ is a multi-ton}\}$
11: {Peeling}
12: **while** $|\mathcal{S}| > 0$ **do**
13:     **for** each $(c,\mathbf{j}) \in \mathcal{S}$ **do**
14:         Recover $\mathbf{k}$ and $F[\mathbf{k}]$ from $U_{c,p}[\mathbf{j}]$
15:         $F \leftarrow F \cup (\mathbf{k}, F[\mathbf{k}])$
16:         **for** each $c' \in [C]$ **do**
17:             $\mathbf{j}' \leftarrow \mathbf{M}_{c'}^T \mathbf{k}$
18:             $U_{c',p}[\mathbf{j}'] \leftarrow U_{c',p}[\mathbf{j}'] - F[\mathbf{k}] \omega^{\langle \mathbf{d}_{c',p}, \mathbf{k} \rangle}$
19:         **end for**
20:         Update $\mathcal{S}, L$ accordingly
21:     **end for**
22: **end while**
23: **return** $F$ {Fourier transform estimate}

---

**Algorithm 2** SHAP zero: Computing SHAP Values and Faith-Shap Interactions (Per Query Sequnce)

---

**Require:** Input query sequences $\{\mathbf{x}_i\}_{i\in[Q]}$, Fourier transform estimate $F$ with maximum order $\ell$
 1: **for** each input query sequence $\mathbf{x}_i$ **do**
 2:     {Compute Möbius transform around $\mathbf{x}_i$}
 3:     **for** each $\mathbf{k} \in \mathbb{Z}_q^n$ **do**

 4:     $$M_{\mathbf{x}_i}[\mathbf{k}] = \sum_{\mathbf{y} \in \mathbb{Z}_q^n} F[\mathbf{y}] \, \omega^{\langle \mathbf{x}_i, \mathbf{y} \rangle} \left( \sum_{\mathbf{m} \leq \mathbf{k}} (-1)^{\|\mathbf{k}-\mathbf{m}\|_0} \omega^{\langle \mathbf{m}, \mathbf{y} \rangle} \right)$$

 5:     **end for**
 6:     {Map Möbius transform to SHAP values}
 7:     **for** each feature $j = 1, \dots, n$ **do**
 8:         $I_{\mathbf{x}_i}^{SV}(j) = - \sum_{\mathbf{k}:k_j>0} \frac{1}{\|\mathbf{k}\|_0 q^{\|\mathbf{k}\|_0}} M_{\mathbf{x}_i}[\mathbf{k}]$
 9:     **end for**
10:     $\phi_i \leftarrow [I_{\mathbf{x}_i}^{SV}(1), \dots, I_{\mathbf{x}_i}^{SV}(n)]$
11:     {Map Möbius transform to Faith-Shap interactions}
12:     **for** each interaction set $T \in T_\ell$ **do**
13:         $I_{\mathbf{x}_i}^{FSI}(T) = (-1)^{|T|} \sum_{\mathbf{k}:\mathbf{k}_T > \mathbf{0}_{|T|}} \frac{1}{q^{\|\mathbf{k}\|_0}} M_{\mathbf{x}_i}[\mathbf{k}]$
14:     **end for**
15:     $\mathbf{\Phi}_i \leftarrow [I_{\mathbf{x}_i}^{FSI}(T), \quad \forall T \in T_\ell]$
16: **end for**
17: **return** $\{\phi_i, \mathbf{\Phi}_i\}_{i\in[Q]}$

---

## B.1  Shapley Explanations, Möbius Transform, and Fourier Transform in SHAP zero

The connection between Shapley explanations, the Möbius transform, and the sparse Fourier transform lies in how we represent and recover feature interactions in complex models. SHAP values attribute

a model's prediction to individual features and their combinations by decomposing the function into additive contributions over subsets. Similarly, Faith-Shap interactions extend SHAP values over interactions of features. These decompositions align with the Möbius transform, a mathematical framework that captures how a function defined over feature subsets can be expressed in terms of unique contributions from each subset. However, in practice, the exact computation of this decomposition is infeasible for high-dimensional problems due to its exponential computational complexity. The sparse Fourier transform offers a computational shortcut by exploiting the empirical sparsity of interactions in biological sequence models.

The only approximation in SHAP zero is computing the sparse Fourier transform. Given an exact Fourier transform and infinite compute power, one could compute the Möbius transform and, consequently, SHAP values and Faith-Shap interactions, exactly. Thus, the quality of Shapley estimates is heavily dependent on how well the Fourier coefficients approximate the model. To facilitate the easy usage of SHAP zero, we have released a pip package (https://github.com/amirgroup-codes/shap-zero) that automatically tunes the parameters of SHAP zero and provides the best estimate of the Fourier transform given a user's computational budget.

## C  Computing the Fourier Transform

### C.1  Example of a Sparse Fourier Tranform

The key to SHAP zero lies in the assumption that the model is $s$-sparse in the Fourier domain. To illustrate an example of sparsity in Fourier, consider the following function, defined over $n = 3$ sites for a $q = 4$ alphabet, modeled using Equation (3):

$$f(\mathbf{m}) = (5 + 2j)\omega^{\langle \mathbf{m}, [0,1,0] \rangle} - (3 + j)\omega^{\langle \mathbf{m}, [1,2,0] \rangle} + (1 + j)\omega^{\langle \mathbf{m}, [1,0,3] \rangle}.$$

$f$ contains the Fourier coefficients $F[[0, 1, 0]] = 5 + 2j$, $F[[1, 2, 0]] = -(3 + j)$, and $F[[1, 0, 3]] = 1 + j$. The Fourier transform is sparse with $s = 3$ non-zero coefficients out of $q^n = 4^3 = 64$ possible coefficients. Additionally, the Fourier transform also has a maximum interaction order of $\ell = 2$ ($\|[1, 2, 0]\|_0 = \|[1, 0, 3]\|_0 = 2$).

### C.2  Estimating the Sparse Fourier Transform

In this section, we add more details about finding the sparse Fourier transform. The procedure SHAP zero takes can be broken down into three steps: 1) Subsampling and aliasing, 2) Bin detection, and 3) Peeling.

**Subsampling and aliasing.** From Equation (4), we note that the coefficients $U_{c,p}[\mathbf{j}]$ are aliased versions of the coefficients $F[\mathbf{k}]$. We denote $\mathbf{U}_c[\mathbf{j}]$ as the grouping of subsampled Fourier coefficients $\mathbf{U}_c[\mathbf{j}] = [U_{c,1}[\mathbf{j}], \ldots, U_{c,P}[\mathbf{j}]]$, with their respective offsets also grouped up into the matrix $\mathbf{D}_c \in \mathbb{Z}_q^{P \times n}$. We rewrite Equation (4) as:

$$\mathbf{U}_c[\mathbf{j}] = \sum_{\mathbf{k}: \, \mathbf{M}_c^T \mathbf{k} = \mathbf{j}} F[\mathbf{k}] \omega^{\mathbf{D}_c \mathbf{k}}. \tag{11}$$

We additionally denote $\mathbf{s}_{c,\mathbf{k}} = \omega^{\mathbf{D}_c \mathbf{k}}$, which allows us to re-write the above equation as:

$$\mathbf{U}_c[\mathbf{j}] = \sum_{\mathbf{k}: \, \mathbf{M}_c^T \mathbf{k} = \mathbf{j}} F[\mathbf{k}] \mathbf{s}_{c,\mathbf{k}}. \tag{12}$$

**Noiseless Bin Detection.** After aliasing $f$, we aim to recover the Fourier coefficients using a bin detection procedure, where we aim to identify which $\mathbf{U}_c[\mathbf{j}]$ contains only one Fourier coefficient. For simplicity, we break this into two sections: noiseless (when the Fourier transform is perfectly $s$-sparse) and noisy (when the Fourier transform is approximately $s$-sparse) bin detection.

In the noiseless case, suppose that for each subsampling group $c$, we choose $\mathbf{M}_c$ to be a partial-identity matrix structure of the form:

$$\mathbf{M}_c = [\mathbf{0}_{b \times b(c-1)}, \mathbf{I}_{b \times b}, \mathbf{0}_{b \times (n-cb)}]^T. \tag{13}$$

Fig. 1a shows an example of the aliasing pattern with the partial-identity matrix structure. Additionally, we choose $P = n$ (the number of offsets) and $\mathbf{D}_c = \mathbf{I}_{n \times n}$. We also choose a fixed delay $\mathbf{d}_{c,0} = \mathbf{0}_n$,

where $\mathbf{0}_n$ is the vector of all 0's of length $n$. Consequently, $U_{c,0}[\mathbf{j}] = \sum_{\mathbf{k}:\mathbf{M}_c^T\mathbf{k}=\mathbf{j}} F[\mathbf{k}]$. We then use $U_{c,0}[\mathbf{j}]$ to identify $\mathbf{U}_c[\mathbf{j}]$ as a *zero-ton*, *singleton*, or *multi-ton* bin based on the following criteria:

- **Zero-ton verification:** $\mathbf{U}_c[\mathbf{j}]$ is a zero-ton if $F[\mathbf{k}] = 0$ for vectors $\mathbf{k}$ such that $\mathbf{M}_c^T\mathbf{k} = \mathbf{j}$ (the summation condition in Equation (12). In the noiseless case, this is true when $U_{c,p}[\mathbf{j}] = 0$ for all $p = 1, \ldots, P$.

- **Singleton verification:** $\mathbf{U}_c[\mathbf{j}]$ is a singleton if there exists only one $\mathbf{k}$ where $F[\mathbf{k}] \neq 0$, given $\mathbf{M}_c^T\mathbf{k} = \mathbf{j}$. In the noiseless case, this implies that $|U_{c,1}[\mathbf{j}]| = |U_{c,2}[\mathbf{j}]| = \ldots = |U_{c,P}[\mathbf{j}]|$. This allows us to denote $\mathbf{U}_c[\mathbf{j}]$ as a singleton if the following condition is met:

$$\left| \frac{U_{c,p}[\mathbf{j}]}{U_{c,0}[\mathbf{j}]} \right| = 1, \quad p = 1, 2, \ldots, P.$$

- **Multi-ton verification:** $\mathbf{U}_c[\mathbf{j}]$ is a multi-ton if there exists more than one $\mathbf{k}$ where $F[\mathbf{k}] \neq 0$, given $\mathbf{M}_c^T\mathbf{k} = \mathbf{j}$. In the noiseless case, this means that:

$$\left| \frac{U_{c,p}[\mathbf{j}]}{U_{c,0}[\mathbf{j}]} \right| \neq 1, \quad p = 1, 2, \ldots, P.$$

If a singleton is found, we know that $U_{c,0}[\mathbf{j}] = F[\mathbf{k}]$. The only unknown is what the vector $\mathbf{k}$ is. By exploiting the identity structure of $\mathbf{D}_c$, we know $\mathbf{k}$ must satisfy:

$$\begin{bmatrix} \arg_q[U_{c,1}[\mathbf{j}]/U_{c,0}[\mathbf{j}]] \\ \vdots \\ \arg_q[U_{c,P}[\mathbf{j}]/U_{c,0}[\mathbf{j}]] \end{bmatrix} = \mathbf{D}_c\mathbf{k}. \tag{14}$$

Here, $\arg_q \colon \mathbb{C} \to \mathbb{Z}_q$ is the $q$-quantization of the argument of a complex number:

$$\arg_q(z) := \left\lfloor \frac{q}{2\pi} \arg(z e^{\frac{j\pi}{q}}) \right\rfloor. \tag{15}$$

Therefore, after obtaining the vector $\mathbf{k}$ from the above equation, we know $F[\mathbf{k}] = U_{c,0}[\mathbf{j}]$.

**Noisy Bin Detection.** In the noisy case, we cannot assume that most coefficients are exactly zero. We model this problem as one where the subsampled Fourier coefficients are corrupted by noise:

$$U_{c,p}[\mathbf{j}] = \sum_{\mathbf{k}:\, \mathbf{M}_c^T\mathbf{k}=\mathbf{j}} F[\mathbf{k}]\,\omega^{\langle \mathbf{d}_{c,p}, \mathbf{k} \rangle} + W_{c,p}[\mathbf{j}], \tag{16}$$

where $W_{c,p}[\mathbf{j}]$ is Gaussian noise with zero mean and variance $\nu^2$, such that $\nu^2 = \sigma^2/B$.

To recover the sparse Fourier transform, we relax the constraints made to $\mathbf{M}_c$ and $\mathbf{D}_c$ in the noiseless case. We let $\mathbf{M}_c$ be a random matrix in $\mathbb{Z}_q^{n \times b}$, and given the hyperparameter $P_1$, we let $P = P_1 n$ and $\mathbf{D}_c$ be a random matrix in $\mathbb{Z}_q^{P \times n}$. In the presence of noise $\sigma^2$, which is a hyperparameter in practice, for some $\gamma \in (0, 1)$, we do bin detection using the following criteria:

- **Zero-ton verification:** $\mathbf{U}_c[\mathbf{j}]$ is a zero-ton if $\frac{1}{P}\|\mathbf{U}_c[\mathbf{j}]\|^2 \leq (1 + \gamma)\nu^2$.

- **Singleton verification:** After ruling out zero-tons, obtain an estimate of $\mathbf{k}$, which we denote as $\hat{\mathbf{k}}$. Then, estimate $F[\hat{\mathbf{k}}]$ as $\mathbf{s}_{c,\hat{\mathbf{k}}}^T \mathbf{U}_c[\mathbf{j}]/P$. We can then check if $\mathbf{U}_c[\mathbf{j}]$ is a singleton if:

$$\frac{1}{P}\|\mathbf{U}_c[\mathbf{j}] - F[\hat{\mathbf{k}}]\mathbf{s}_{c,\hat{\mathbf{k}}}\|^2 \leq (1 + \gamma)\nu^2.$$

- **Multi-ton verification:** If $\mathbf{U}_c[\mathbf{j}]$ is neither a zero-ton nor a singleton, then $\mathbf{U}_c[\mathbf{j}]$ is a multi-ton.

SHAP zero obtains an estimate of $\mathbf{k}$ for singleton detection based on repetition coding. Given our $P_1$ random offsets $\mathbf{d}_{p \in [P_1]}$ such that $P = P_1 n$, we construct perturbed versions of each offset by modulating with each column of the identity matrix such that we have $n$ offsets $\mathbf{d}_{p,r}$:

$$\mathbf{d}_{p,r} = \mathbf{d}_p \oplus_q \mathbf{e}_r, \forall p \in [P_1], \quad \forall r \in n. \tag{17}$$

This allows us to employ a majority test to estimate the $r^{\text{th}}$ value of $\hat{\mathbf{k}}$:

$$\hat{k}_r = \arg\max_{a \in \mathbb{Z}_q} \sum_{p \in [P_1]} \mathbb{1}\{a = \arg_q[U_{c,p,r}/U_{c,p}]\}, \tag{18}$$

where $\mathbb{1}$ is the indicator function. SHAP zero uses noisy bin detection to recover the sparse Fourier transform. Details about how to run the sparse Fourier transform in practice can be found in Appendix G.1.

**Peeling.** After determining the bin type, we apply a peeling decoder to recover additional singletons. The problem is modeled as a sparse bipartite graph: each $F[\mathbf{k}]$ corresponds to a variable node, each $\mathbf{U}_c[\mathbf{j}]$ is a check node, and an edge connects $F[\mathbf{k}]$ and $\mathbf{U}_c[\mathbf{j}]$ when $\mathbf{M}_c^T \mathbf{k} = \mathbf{j}$.

Upon identifying a singleton, for every subsampling group $c$, we subtract $F[\mathbf{k}]$ from the check node $\mathbf{U}_c[\mathbf{M}_c^T \mathbf{k}]$. This effectively peels an edge off the bipartite graph. We repeat this process over all subsampled Fourier coefficients, iteratively peeling singletons until no further singletons remain.

After peeling can no longer be done, SHAP zero recovers all Fourier coefficients with a sample complexity of $\mathcal{O}(sn^2)$ and a computational complexity of $\mathcal{O}(sn^3)$. We refer readers to [40] for a comprehensive theoretical analysis.

### C.3 Sparsity and Interaction Order in Encoding $q$-ary Functions

Our theoretical analysis shows that the performance of SHAP zero depends on $s$ and $\ell$. A natural question arises: does the choice of encoding scheme in $q$-ary functions (e.g., in DNA, $A = 0, C = 1,$ $T = 2, G = 3$ versus $C = 0, T = 1, G = 2, A = 3$) affect $s$ or $\ell$?

We emphasize that SHAP zero is *robust* to the choice of encoding scheme. Both $s$ and $\ell$ are intrinsic to the underlying sequence-function relationship. By the shifting property of the Fourier transform (Equation (21) in Appendix E.1), changing the encoding scheme will only change the magnitude of non-zero Fourier coefficients, leaving $s$ and $\ell$ untouched. Consequently, the theoretical guarantees of SHAP zero are unaffected by the encoding scheme.

To validate this empirically, we consider a multilayer perceptron trained on RNA sequences of length $n = 9$ from [63] predicting splicing efficiency [64]. We ran SHAP zero under four encoding schemes (see Table 1) and, following the methodology described in Appendix G.1, computed the $R^2$ when reconstructing model predictions on $10,000$ random sequences using the recovered Fourier coefficients. Although $s$ varied across encoding schemes ($437 \pm 119$), likely due to the model not being exactly sparse, $R^2$ values remained highly consistent ($0.78 \pm 0.01$), and $\ell$ was consistently $4$. These results empirically demonstrate that SHAP zero is robust to the choice of encoding scheme in real-world models.

| Encoding scheme | | | | $s$ | $R^2$ | $\ell$ |
|---|---|---|---|---|---|---|
| $A = 0$ | $C = 1$ | $U = 2$ | $G = 3$ | 567 | 0.80 | 4 |
| $A = 1$ | $C = 2$ | $U = 3$ | $G = 0$ | 326 | 0.77 | 4 |
| $A = 2$ | $C = 3$ | $U = 0$ | $G = 1$ | 345 | 0.78 | 4 |
| $A = 3$ | $C = 0$ | $U = 1$ | $G = 2$ | 508 | 0.77 | 4 |

Table 1: $s$, $R^2$, and $\ell$ across different RNA base encodings when recovering Fourier coefficients with SHAP zero.

## D  Computing the Möbius Transform

### D.1  Set Möbius Transform

From Definition 2.1, each set Möbius transform coefficient $a(v, S)$, where $S \subseteq T$, represents the marginal contribution of the subset $S$, given a value function $v$, in determining the score $v_T(\mathbf{x})$. Reconstructing the original score $v_T(\mathbf{x})$ from all subsets $S \subseteq T$ requires taking a linear sum over all the marginal contributions. For instance, in an $n = 4$ problem, say we want to compute the set Möbius transform of the set $S = \{1, 4\}$. Let $v_\emptyset(\mathbf{x})$ be the output of the value function $v$ when no

inputs are given. The set Möbius transform $a(v, S)$ can be written as:

$$a(v, \{1, 4\}) = (-1)^{|\{1,4\}|-|\{1,4\}|} v_{\{1,4\}}(\mathbf{x}) + (-1)^{|\{1,4\}|-|\{1\}|} v_{\{1\}}(\mathbf{x})$$
$$+ (-1)^{|\{1,4\}|-|\{4\}|} v_{\{4\}}(\mathbf{x}) + (-1)^{|\{1,4\}|-|\emptyset|} v_{\emptyset}(\mathbf{x}).$$

Each exponent raised to the negative one corresponds to the sign of each value function's output. Simplifying each exponent results in the following:

$$a(v, \{1, 4\}) = v_{\{1,4\}}(\mathbf{x}) - v_{\{1\}}(\mathbf{x}) - v_{\{4\}}(\mathbf{x}) + v_{\emptyset}(\mathbf{x}).$$

Now, suppose we want to compute the inverse set Möbius transform of the set $T = \{1, 4\}$. The value function can be written in terms of $a(v, S)$:

$$v_{\{1,4\}}(\mathbf{x}) = a(v, \emptyset) + a(v, \{1\}) + a(v, \{4\}) + a(v, \{1, 4\}).$$

### D.2 $q$-ary Möbius Transform

**Examples of Standard Real Field Operations.** Computing the $q$-ary Möbius transform into the $q$-ary domain involves utilizing standard real field operations over $\mathbb{Z}_q^n \to \mathbb{R}$. To illustrate the subtraction of $\mathbf{k} - \mathbf{m}$, consider the vectors $\mathbf{k} = [1, 0, 0, 2]$ and $\mathbf{m} = [1, 0, 0, 0]$. Then,

$$\|\mathbf{k} - \mathbf{m}\|_0 = \|[1, 0, 0, 2] - [1, 0, 0, 0]\|_0 = \|[0, 0, 0, 2]\|_0 = 1.$$

This is only computable when $\mathbf{m} \le \mathbf{k}$. For example, the vectors $\mathbf{m} = [1, 0, 0, 0]$ and $\mathbf{k} = [1, 0, 0, 2]$ satisfy $\mathbf{m} \le \mathbf{k}$: $m_1 = k_1 = 1$, $m_2 = k_2 = 0$, $m_3 = k_3 = 0$, and $m_4 = 0$. An example of vectors that do not satisfy $\mathbf{m} \le \mathbf{k}$ are $\mathbf{m} = [1, 0, 1, 1]$ and $\mathbf{k} = [1, 0, 0, 2]$. Here, $m_3 = 1 \ne k_3 = 0$ and $m_4 = 1 \ne k_4 = 2$, so $\mathbf{m} \le \mathbf{k}$ is violated.

**Computing the $q$-ary Möbius Transform.** To illustrate computing the $q$-ary Möbius transform from $f$, consider a function where $q = 4$ and $n = 4$, and define the encoding scheme as $A = 0$, $C = 1$, $T = 2$, and $G = 3$. Suppose the query sequence is $\mathbf{x}_i = [2, 1, 0, 3]$ (corresponding to the sequence 'TCAG'). We wish to compute the $q$-ary Möbius transform $M_{\mathbf{x}_i}[\mathbf{k}]$ for $\mathbf{k} = [1, 0, 0, 2]$.

Using the definition from Equation (5), we have:

$$M_{\mathbf{x}_i}[[1, 0, 0, 2]] = \sum_{\mathbf{m} \le [1,0,0,2]} (-1)^{\|[1,0,0,2] - \mathbf{m}\|_0} f\left((\mathbf{m} + \mathbf{x}_i) \bmod 4\right). \tag{19}$$

Expanding the sum over all $\mathbf{m} \le [1, 0, 2, 0]$:

$$M_{\mathbf{x}_i}[[1, 0, 0, 2]] = (-1)^0 f\left(([1, 0, 0, 2] + [2, 1, 0, 3]) \bmod 4\right)$$
$$+ (-1)^1 f\left(([1, 0, 0, 0] + [2, 1, 0, 3]) \bmod 4\right)$$
$$+ (-1)^1 f\left(([0, 0, 0, 2] + [2, 1, 0, 3]) \bmod 4\right)$$
$$+ (-1)^2 f\left(([0, 0, 0, 0] + [2, 1, 0, 3]) \bmod 4\right).$$

Simplifying each term of $f$ results in:

- $[1, 0, 0, 2] + [2, 1, 0, 3] = [3, 1, 0, 5]$; $[3, 1, 0, 5] \bmod 4 = [3, 1, 0, 1]$
- $[1, 0, 0, 0] + [2, 1, 0, 3] = [3, 1, 0, 3]$; $[3, 1, 0, 3] \bmod 4 = [3, 1, 0, 3]$
- $[0, 0, 0, 2] + [2, 1, 0, 3] = [2, 1, 0, 5]$; $[2, 1, 0, 5] \bmod 4 = [2, 1, 0, 1]$
- $[0, 0, 0, 0] + [2, 1, 0, 3] = [2, 1, 0, 3]$; $[2, 1, 0, 3] \bmod 4 = [2, 1, 0, 3]$

This allows us to simplify as follows:

$$M_{\mathbf{x}_i}[[1, 0, 0, 2]] = f([3, 1, 0, 1]) - f([3, 1, 0, 3]) - f([2, 1, 0, 1]) + f([2, 1, 0, 3]).$$

**Computing the Inverse $q$-ary Möbius Transform.** Now, we illustrate computing the inverse $q$-ary Möbius transform to recover $f$ from $M_{\mathbf{x}_i}$. Suppose again the same query sequence $\mathbf{x}_i = [2, 1, 0, 3]$, except we wish to compute $f((\mathbf{m} + \mathbf{x}_i) \bmod 4)$ for $\mathbf{m} = [1, 0, 0, 2]$.

Using the definition from Equation (6), we have:

$$f\left(([1, 0, 0, 2] + \mathbf{x}_i) \bmod 4\right) = \sum_{\mathbf{k} \le [1,0,0,2]} M_{\mathbf{x}_i}[\mathbf{k}]. \tag{20}$$

Expanding the sum over all $\mathbf{k} \leq [1, 0, 0, 2]$:

$$f\left(([1, 0, 0, 2] + [2, 1, 0, 3]) \bmod 4\right) = M_{\mathbf{x}_i}[[0, 0, 0, 0]] + M_{\mathbf{x}_i}[[1, 0, 0, 0]]$$
$$+ M_{\mathbf{x}_i}[[0, 0, 0, 2]] + M_{\mathbf{x}_i}[[1, 0, 0, 2]].$$

This results in the following:

$$f\left([3, 1, 0, 1]\right) = M_{\mathbf{x}_i}[[0, 0, 0, 0]] + M_{\mathbf{x}_i}[[1, 0, 0, 0]] + M_{\mathbf{x}_i}[[0, 0, 0, 2]] + M_{\mathbf{x}_i}[[1, 0, 0, 2]].$$

# E  Proofs

In this section, we provide proofs for the Proposition 3.2, Proposition 3.3, Theorem 3.4, and Theorem 3.5. For simplicity in notation, we drop the subscript $i$ from $\mathbf{x}_i$ and write $\mathbf{x}$.

## E.1  Proof of Proposition 3.2

In the Fourier transform, all alphabets in $q$ are treated as a dictionary that maps an input to a $q$-ary value. However, since the $q$-ary Möbius transform generalizes the $\leq$ operation in $\mathbb{Z}_q^n \to \mathbb{R}$, for each input query sequence $\mathbf{x}$, we utilize the shifting property of the Fourier transform. The shifting property of the Fourier transform states that every Fourier coefficient must be shifted by the sample's $q$-ary encoding using the following equation [40]:

$$\hat{F}[\mathbf{y}] = F[\mathbf{y}]\omega^{\langle \mathbf{x}, \mathbf{y} \rangle}, \quad \forall \mathbf{y} \in \mathbb{Z}_q^n \tag{21}$$

where $\hat{F}[\mathbf{y}]$ refers to the shifted Fourier coefficient, $\mathbf{x}$ refers to the Fourier-encoded query sequence, and $F[\mathbf{y}]$ refers to the original $q$-ary Fourier coefficient. To shift the entire $s$-sparse Fourier transform, we have to loop over all $s$ Fourier coefficients and shift them by the query sequence.

As a consequence of the shifting property of the Fourier transform, we can re-write Equation (3) as:

$$f((\mathbf{m} + \mathbf{x}) \bmod q) = \sum_{\mathbf{y} \in \mathbb{Z}_q^n} \hat{F}[\mathbf{y}]\omega^{\langle \mathbf{m}, \mathbf{y} \rangle}. \tag{22}$$

To illustrate this further, consider the query sequence $[0, 1, 2]$, assuming the encoding pattern where $0 = A$, $1 = C$, $2 = G$, and $3 = T$. The shifting property of the Fourier transform allows us to characterize this same sequence as $[0, 0, 0]$, which satisfies the $\leq$ operation (see Appendix D.2). Generalizing this, we aim to shift the Fourier transform in such a way that our sequence $\mathbf{x}$ is encoded as $\mathbf{0}_n$.

After obtaining $\hat{F}[\mathbf{y}]$, we plug in Equation (22) into Equation (5) to obtain:

$$M_{\mathbf{x}}[\mathbf{k}] = \sum_{\mathbf{m} \leq \mathbf{k}} (-1)^{\|\mathbf{k} - \mathbf{m}\|_0} \left( \sum_{\mathbf{y} \in \mathbb{Z}_q^n} \hat{F}[\mathbf{y}]\omega^{\langle \mathbf{m}, \mathbf{y} \rangle} \right). \tag{23}$$

By pulling $\sum_{\mathbf{y} \in \mathbb{Z}_q^n} \hat{F}[\mathbf{y}]$ out of the summation $\sum_{\mathbf{m} \leq \mathbf{k}}(-1)^{\|\mathbf{k} - \mathbf{m}\|_0}$, we obtain the mapping from the $q$-ary Fourier transform to $M_{\mathbf{x}}[\mathbf{k}]$ as given in Equation (7):

$$M_{\mathbf{x}}[\mathbf{k}] = \sum_{\mathbf{y} \in \mathbb{Z}_q^n} \hat{F}[\mathbf{y}] \left( \sum_{\mathbf{m} \leq \mathbf{k}} (-1)^{\|\mathbf{k} - \mathbf{m}\|_0} \omega^{\langle \mathbf{m}, \mathbf{y} \rangle} \right)$$
$$= \sum_{\mathbf{y} \in \mathbb{Z}_q^n} F[\mathbf{y}]\omega^{\langle \mathbf{x}, \mathbf{y} \rangle} \left( \sum_{\mathbf{m} \leq \mathbf{k}} (-1)^{\|\mathbf{k} - \mathbf{m}\|_0} \omega^{\langle \mathbf{m}, \mathbf{y} \rangle} \right), \quad \forall \mathbf{k} \in \mathbb{Z}_q^n.$$

Equation (7) details how to convert the shifted Fourier transform to $M_{\mathbf{x}}[\mathbf{k}]$ for a single $\mathbf{k}$ (a single $q$-ary Möbius coefficient). If no assumptions are made, the complexity of mapping the Fourier transform into every possible $M_{\mathbf{x}}[\mathbf{k}]$ scales with $\mathcal{O}(q^n)$. However, if we assume the Fourier transform is low-order and has a maximum order of $\ell$, we can bound the number of nonzero elements in $M_{\mathbf{x}}[\mathbf{k}]$. This is because Equation (7) will always go to zero when there exists an $i$ such that $y_i = 0$ and $k_i \neq 0$. As an example, consider finding the $q$-ary Möbius transform of a function which is $s = 1$-sparse

in Fourier and let us assume $\hat{F}[[2,1,0]]$ is the only non-zero Fourier coefficient. In this setup, one example of $\mathbf{k}$ for which $M_{\mathbf{x}}[\mathbf{k}]$ is zero is $\mathbf{k} = [0,1,1]$ since it satisfies the condition that $y_3 = 0$ and $k_3 \neq 0$. Computing Equation (7), we get:

$$M_{\mathbf{x}}[\mathbf{k}] = \hat{F}[[2,1,0]]\Bigg( (-1)^{\|[0,1,1]-[0,0,0]\|_0}\omega^{\langle[0,0,0],[2,1,0]\rangle} + (-1)^{\|[0,1,1]-[0,1,0]\|_0}\omega^{\langle[0,1,0],[2,1,0]\rangle}$$

$$+ (-1)^{\|[0,1,1]-[0,0,1]\|_0}\omega^{\langle[0,0,1],[2,1,0]\rangle} + (-1)^{\|[0,1,1]-[0,1,1]\|_0}\omega^{\langle[0,1,1],[2,1,0]\rangle} \Bigg).$$

From here, we can simplify to the following:

$$M_{\mathbf{x}}[\mathbf{k}] = \hat{F}[[2,1,0]]\Bigg( (-1)^2\omega^0 + (-1)^1\omega^1 + (-1)^1\omega^0 + (-1)^0\omega^1 \Bigg) = 0.$$

As a consequence, now we can bound the number of nonzero elements in $M_{\mathbf{x}}[\mathbf{k}]$ by counting the number of vectors $\mathbf{k}$ for which there is no $i$ such that $y_i = 0$ and $k_i \neq 0$. If we assume that the Fourier transform is low-order and bounded by $\ell$, then we can bound it by $\mathcal{O}(q^\ell)$ instead of $\mathcal{O}(q^n)$.

Generalizing this example, if there exists an $i$ such that $y_i = 0$ and $k_i \neq 0$, this allows us to rewrite Equation (7) as the following:

$$M_{\mathbf{x}}[\mathbf{k}] = \sum_{\mathbf{y} \in \mathbb{Z}_q^n} \hat{F}[\mathbf{y}]\Bigg( \sum_{\substack{\mathbf{m} \leq \mathbf{k} \\ m_i = 0}} (-1)^{\|\mathbf{k}-\mathbf{m}\|_0}\omega^{\langle\mathbf{m},\mathbf{y}\rangle} + (-1)^{\|\mathbf{k}-\mathbf{m}^+\|_0}\omega^{\langle\mathbf{m}^+,\mathbf{y}\rangle} \Bigg),$$

where $\mathbf{m}^+$ is equal to $\mathbf{m}$ everywhere except at location $i$ and $\mathbf{m}^+{}_i = k_i$. Since $y_i = 0$, $\langle\mathbf{m},\mathbf{y}\rangle = \langle\mathbf{m}^+,\mathbf{y}\rangle$. Additionally, we can rewrite $\|\mathbf{k}-\mathbf{m}\|_0$ as $\|\mathbf{k}-\mathbf{m}^+\|_0 + 1$, since we know that the only difference between $\mathbf{m}^+$ and $\mathbf{m}$ is a single nonzero value at position $i$. Therefore, we can say that:

$$M_{\mathbf{x}}[\mathbf{k}] = \sum_{\mathbf{y} \in \mathbb{Z}_q^n} \hat{F}[\mathbf{y}]\Bigg( \sum_{\substack{\mathbf{m} \leq \mathbf{k} \\ m_i = 0}} (-1)^{\|\mathbf{k}-\mathbf{m}\|_0}\omega^{\langle\mathbf{m},\mathbf{y}\rangle} + (-1)^{\|\mathbf{k}-\mathbf{m}\|_0+1}\omega^{\langle\mathbf{m},\mathbf{y}\rangle} \Bigg) = 0.$$

We now move to analyze the computational complexity of converting from the Fourier to the $q$-ary Möbius transform. First, shifting all the Fourier coefficients scales with $\mathcal{O}(s)$, since we have to loop through all $s$ Fourier coefficients once. The time complexity of doing the conversion using Equation (7) depends on the distribution of the Fourier coefficient interaction orders. In the worst-case scenario, we can assume that all Fourier coefficients are all $\ell$-th order. We break up the problem into two categories: the computational complexity of computing the most expensive $q$-ary Möbius coefficient and analyzing the total amount of $q$-ary Möbius coefficients created. From Equation (7), the most expensive $q$-ary Möbius coefficient to compute will be an $\ell$-th order Möbius coefficient due to summing over $2^\ell$ vectors, which is the total amount of vectors encompassed in $\mathbf{m} \leq \mathbf{k}$; this single $q$-ary Möbius coefficient requires $s$ summations over $2^\ell$ vectors, which scales with computational complexity $\mathcal{O}(s2^\ell)$. Additionally, if all $s$ Fourier coefficients are $\ell$-th order, there are at most $sq^\ell$ possible $q$-ary Möbius coefficients to loop through. Therefore, the overall computational complexity of converting from the Fourier to the $q$-ary Möbius transform is $\mathcal{O}(s + s2^\ell \times sq^\ell) \approx \mathcal{O}(s^2(2q)^\ell)$.

### E.2 Proof of Proposition 3.3

To prove Proposition 3.3, we require using the $q$-ary value function, which we leave here for reference:

$$v_T(\mathbf{x}) = \frac{1}{q^{|\bar{T}|}} \sum_{\mathbf{m}:\mathbf{m}_{\bar{T}} \in \mathbb{Z}_q^{|\bar{T}|}, \mathbf{m}_T = \mathbf{x}_T} f(\mathbf{m}), \tag{24}$$

We plug in the inverse $q$-ary Möbius transform from Equation (6) into the $q$-ary value function. By using $M_{\mathbf{x}}[\mathbf{k}]$, this means that we shift $\mathbf{m}$ such that $\mathbf{m}_T = \mathbf{0}_{|T|}$. Here, value 0 is assigned to subset indices $T$, which indexes the subset of the original inputs $\mathbf{x}_T$. Over the missing inputs $\mathbf{x}_{\bar{T}}$ and their

respective indices $\bar{T}$, the vector $\mathbf{m}_{\bar{T}} \in \mathbb{Z}_{\mathbf{q}_{\bar{T}}}$ is used, by definition of the value function:

$$v_T(\mathbf{x}) = \frac{1}{q^{|\bar{T}|}} \sum_{\substack{\mathbf{m}:\mathbf{m}_{\bar{T}} \in \mathbb{Z}_q^{|\bar{T}|} \\ \mathbf{m}_T = \mathbf{x}_T}} f(\mathbf{m}) = \frac{1}{q^{|\bar{T}|}} \sum_{\substack{\mathbf{m}:\mathbf{m}_{\bar{T}} \in \mathbb{Z}_q^{|\bar{T}|} \\ \mathbf{m}_T = \mathbf{0}_T}} f((\mathbf{m} + \mathbf{x}) \bmod q)$$

$$= \frac{1}{q^{|\bar{T}|}} \sum_{\substack{\mathbf{m}:\mathbf{m}_{\bar{T}} \in \mathbb{Z}_q^{|\bar{T}|} \\ \mathbf{m}_T = \mathbf{0}_{|T|}}} \left( \sum_{\mathbf{k} \leq \mathbf{m}} M_{\mathbf{x}}[\mathbf{k}] \right).$$

Due to the recursive nature of the $q$-ary Möbius transform, each $q$-ary Möbius coefficient is looped through $q^{|\bar{T}| - \|\mathbf{m}\|_0}$ times. Therefore, we can simplify as follows:

$$v_T(\mathbf{x}) = \frac{1}{q^{|\bar{T}|}} \sum_{\substack{\mathbf{m}:\mathbf{m}_{\bar{T}} \in \mathbb{Z}_q^{|\bar{T}|} \\ \mathbf{m}_T = \mathbf{0}_{|T|}}} q^{|\bar{T}| - \|\mathbf{m}\|_0} M_{\mathbf{x}}[\mathbf{m}].$$

We can pull the $q^{|\bar{T}|}$ out of the summation, which will allow us to cancel the $q^{|\bar{T}|}$ term in the denominator. This leaves us with the following:

$$v_T(\mathbf{x}) = \sum_{\substack{\mathbf{m}:\mathbf{m}_{\bar{T}} \in \mathbb{Z}_q^{|\bar{T}|} \\ \mathbf{m}_T = \mathbf{0}_{|T|}}} \frac{1}{q^{\|\mathbf{m}\|_0}} M_{\mathbf{x}}[\mathbf{m}]. \tag{25}$$

Plugging Equation (25) into the forward set Möbius transform equation from Definition 2.1 returns an equation for the set Möbius transform in terms of the $q$-ary Möbius transform:

$$a(v, S) = \sum_{T \subseteq S} \left( (-1)^{|S| - |T|} \sum_{\substack{\mathbf{m}:\mathbf{m}_{\bar{T}} \in \mathbb{Z}_q^{|\bar{T}|} \\ \mathbf{m}_T = \mathbf{0}_{|T|}}} \frac{1}{q^{\|\mathbf{m}\|_0}} M_{\mathbf{x}}[\mathbf{m}] \right). \tag{26}$$

Although Equation (26) loops only over subsets $T \subseteq S$, all possible $q$-ary Möbius coefficients are accounted for, since $\mathbf{m}$ is nonzero at coordinates given by $\bar{T}$. We rewrite Equation (26) such that we loop over all possible $q$-ary Möbius coefficients, given by the subsets $\bar{L} \subseteq D$:

$$a(v, S) = \sum_{\bar{L} \subseteq D} \left( \sum_{\substack{T:T \subseteq S \\ \text{if } \bar{L} \subseteq \bar{T}}} (-1)^{|S| - |T|} \left( \frac{1}{q^{|\bar{L}|}} \sum_{\substack{\mathbf{m}:\forall i \in \bar{L}, m_i > 0 \\ \mathbf{m}_L = \mathbf{0}_{|L|}}} M_{\mathbf{x}}[\mathbf{m}] \right) \right).$$

Notably, we are looping through $T \subseteq S$ over the set $T$ in order to compute $(-1)^{|S| - |T|}$, but our if condition is with respect to $\bar{T}$. For every $T$, $\bar{T}$ is implicitly defined already, since $\bar{T}$ is the complement of $T$.

We look to simplify the outer summation. Consider the relationship between an arbitrary $\bar{L}$ given the condition $\bar{L} \subseteq \bar{T}$. $\bar{L} \subseteq \bar{T}$ is only true when $\bar{L} \cap T = \emptyset$ based on the way $\bar{T}$ is implicitly defined. Since $T \subseteq S$, the total number of features that can be in $T$ that can satisfy $\bar{L} \cap T = \emptyset$ is $|S \setminus \bar{L}|$. Therefore, the amount of $T$ subsets that will loop through $T \subseteq S$ is equal to $2^{|S \setminus \bar{L}|}$ for an arbitrary $\bar{L}$. From here, we will cancel out terms by noting that we must loop through the alternating series $2^{|S \setminus \bar{L}|}$ times. We divide $\bar{L}$ into two cases: when $S \subseteq \bar{L}$ and when $S \not\subseteq \bar{L}$. When $S \not\subseteq \bar{L}$, $|S \setminus \bar{L}| > 0$, which means we loop through the alternating series an even amount of times, canceling out any $q$-ary Möbius coefficients that have $|\bar{L}|$ nonzero values. However, when $S \subseteq \bar{L}$, $|S \setminus \bar{L}| = 0$. Therefore, we can rewrite the outer summation to be over all possible $\bar{L}$'s where $S \subseteq \bar{L}$:

$$a(v, S) = \sum_{\bar{L}:S \subseteq \bar{L}} \left( \sum_{\substack{T:T \subseteq S \\ \text{if } \bar{L} \cap T = \emptyset}} (-1)^{|S| - |T|} \left( \frac{1}{q^{|\bar{L}|}} \sum_{\substack{\mathbf{m}:\forall i \in \bar{L}, m_i > 0 \\ \mathbf{m}_L = \mathbf{0}_{|L|}}} M_{\mathbf{x}}[\mathbf{m}] \right) \right).$$

Now, given $S \subseteq \bar{L}$, the only way $T \subseteq S$ and $\bar{L} \cap T = \emptyset$ can be satisfied is when $T = \emptyset$. $T$ cannot contain any features in its set because it will violate $\bar{L} \cap T = \emptyset$. To illustrate this, consider an $n = 3$

problem. Let $S = \{1, 2\}$ and $\bar{L} = \{1, 2, 3\}$. The possible subsets $T \subseteq S$ include $\emptyset$, $\{1\}$, $\{2\}$, and $\{1, 2\}$. For the subsets $\{1\}$ and $\{2\}$, $\{1\} \cap \{1, 2, 3\} = \{1\} \neq \emptyset$ and $\{2\} \cap \{1, 2, 3\} = \{2\} \neq \emptyset$. The only case where $\bar{L} \cap T = \emptyset$ is when $T = \emptyset$, which means $|T| = 0$.

Setting $|T| = 0$ and the outer summation to loop over all possible $\bar{L}$'s where $S \subseteq \bar{L}$, we obtain the following expression:

$$a(v, S) = \sum_{\bar{L}:S \subseteq \bar{L}} \left( (-1)^{|S|} \left( \frac{1}{q^{|\bar{L}|}} \sum_{\substack{\mathbf{m}: \forall i \in \bar{L}, m_i > 0 \\ \mathbf{m}_L = \mathbf{0}_{|L|}}} M_{\mathbf{x}}[\mathbf{m}] \right) \right).$$

Pulling $(-1)^{|S|}$ out of the summation, we obtain an expression to convert from the $q$-ary Möbius transform to set Möbius transform:

$$a(v, S) = (-1)^{|S|} \sum_{\bar{L}:S \subseteq \bar{L}} \left( \frac{1}{q^{|\bar{L}|}} \sum_{\substack{\mathbf{m}: \forall i \in \bar{L}, m_i > 0 \\ \mathbf{m}_L = \mathbf{0}_{|L|}}} M_{\mathbf{x}}[\mathbf{m}] \right). \tag{27}$$

This can equivalently be written in vector notation as:

$$a(v, S) = (-1)^{|S|} \sum_{\mathbf{k}:\mathbf{k}_S > \mathbf{0}_{|S|}} \frac{1}{q^{\|\mathbf{k}\|_0}} M_{\mathbf{x}}[\mathbf{k}]. \tag{28}$$

### E.3 Proof of Theorem 3.4

With a conversion from the $q$-ary Möbius transform to the set Möbius transform, we can now derive the SHAP value formula with respect to the $q$-ary Möbius transform. To do this, we will need to use the following Lemmas.

**Lemma E.1.** *For all $n \geq 0$ and $k \geq 0$, where $n$ and $k$ are integers:*

$$\binom{n-1}{k-1} = \frac{k}{n} \binom{n}{k}.$$

*Proof.* Expanding the right-hand side results in the following: $\binom{n-1}{k-1} = \frac{k}{n} \binom{n}{k} = \frac{k \cdot n!}{n \cdot k! \cdot (n-k)!} = \frac{(n-1)!}{(k-1)! \cdot (n-k)!} = \binom{n-1}{k-1}$. $\qquad\square$

**Lemma E.2.** *For all $n \geq 1$, where $n$ is an integer:*

$$\sum_{k=1}^{n} (-1)^k \binom{n}{k} = -1.$$

*Proof.* We require a special case of the binomial theorem. The binomial theorem states that, for any number $x$ and $y$, and given an integer $n \geq 0$: $(x + y)^n = \sum_{k=0}^{n} \binom{n}{k} x^k y^{n-k}$. Plugging in $y = 1$ and $x = -1$ leads to the following simplification: $\sum_{k=0}^{n} (-1)^k \binom{n}{k} = 0$. We pull out the summation term where $k = 0$, and instead loop the summation over from $k = 1$ to $n$: $1 + \sum_{k=1}^{n} (-1)^k \binom{n}{k} = 0$. Finally, isolating the summation term leaves us with Lemma E.2. $\sum_{k=1}^{n} (-1)^k \binom{n}{k} = -1$. $\qquad\square$

Our derivation will require the classic SHAP value equation, which we leave below. Since we drop the $i$ subscript from $\mathbf{x}_i$, we write $I_{\mathbf{x}}^{SV}(i)$ instead of $I_{\mathbf{x}_i}^{SV}(j)$:

$$I_{\mathbf{x}}^{SV}(i) = \sum_{T \subseteq D \setminus \{i\}} \frac{|T|! \, (|D| - |T| - 1)!}{|D|!} \left[ v_{T \cup \{i\}}(\mathbf{x}) - v_T(\mathbf{x}) \right] \tag{29}$$

To begin our derivation, we plug in Equation (27) into Equation (29):

$$I_{\mathbf{x}}^{SV}(i) = \sum_{T \subseteq D \setminus \{i\}} \frac{1}{|T \cup \{i\}|} \left( (-1)^{|T \cup \{i\}|} \sum_{\bar{L}:T \cup \{i\} \subseteq \bar{L}} \left( \frac{1}{q^{|\bar{L}|}} \sum_{\substack{\mathbf{m}: \forall j \in \bar{L}, m_j > 0 \\ \mathbf{m}_L = \mathbf{0}_{|L|}}} M_{\mathbf{x}}[\mathbf{m}] \right) \right). \tag{30}$$

Summing up over all sets $T \subseteq D \setminus \{i\}$ results in repeatedly looping over the same $q$-ary Möbius coefficients several times. We re-write Equation (30) to sum over all $q$-ary Möbius coefficient sets $\bar{L} \subseteq D \setminus \{i\}$, which allows us to loop over all unique $q$-ary Möbius coefficients as we keep track of how many times the alternating series gets summed:

$$I_{\mathbf{x}}^{SV}(i) = \sum_{\bar{L} \subseteq D \setminus \{i\}} \frac{1}{q^{|\bar{L} \cup \{i\}|}} \left( \sum_{\substack{\mathbf{m}: \forall j \in \bar{L} \cup \{i\}, m_j > 0 \\ \mathbf{m}_{L \setminus \{i\}} = \mathbf{0}_{|L \setminus \{i\}|}}} M_{\mathbf{x}}[\mathbf{m}] \left( \sum_{\substack{T \subseteq \bar{L} \cup \{i\} \\ i \in T}} \frac{1}{|T|}(-1)^{|T|} \right) \right). \quad (31)$$

The amount of times to sum through $\sum_{T \subseteq \bar{L} \cup \{i\}, i \in T} \frac{1}{|T|}(-1)^{|T|}$ depends on $|T|$. For example, consider an $n = 3$ problem, where $i = 1$ and $\bar{L} = \{2, 3\}$. There is only one possible set where $|T| = 1$, $\{1\}$, because $i \in T$. However, there are two possible sets where $|T| = 2$: $\{1, 2\}$ and $\{1, 3\}$. Generalizing this, the amount of possible sets for an arbitrary $|T|$ that satisfy $i \in T$ is equivalent to $\binom{|\bar{L} \cup \{i\}| - 1}{|T| - 1}$. Using this, we re-write Equation (31) by summing over all possibilities of $|T|$:

$$I_{\mathbf{x}}^{SV}(i) = \sum_{\bar{L} \subseteq D \setminus \{i\}} \frac{1}{q^{|\bar{L} \cup \{i\}|}} \left( \sum_{\substack{\mathbf{m}: \forall j \in \bar{L} \cup \{i\}, m_j > 0 \\ \mathbf{m}_{L \setminus \{i\}} = \mathbf{0}_{|L \setminus \{i\}|}}} M_{\mathbf{x}}[\mathbf{m}] \left( \sum_{k=1}^{|\bar{L} \cup \{i\}|} \binom{|\bar{L} \cup \{i\}| - 1}{k - 1} \frac{1}{k}(-1)^k \right) \right).$$

We can apply Lemmas E.1 and E.2 to arrive at the following:

$$I_{\mathbf{x}}^{SV}(i) = \sum_{\bar{L} \subseteq D \setminus \{i\}} \frac{1}{q^{|\bar{L} \cup \{i\}|}} \left( \sum_{\substack{\mathbf{m}: \forall j \in \bar{L} \cup \{i\}, m_j > 0 \\ \mathbf{m}_{L \setminus \{i\}} = \mathbf{0}_{|L \setminus \{i\}|}}} M_{\mathbf{x}}[\mathbf{m}] (\frac{1}{|\bar{L} \cup \{i\}|}) \sum_{k=1}^{|\bar{L} \cup \{i\}|} \binom{|\bar{L} \cup \{i\}|}{k}(-1)^k \right).$$

$$I_{\mathbf{x}}^{SV}(i) = \sum_{\bar{L} \subseteq D \setminus \{i\}} \frac{1}{q^{|\bar{L} \cup \{i\}|}} \left( \sum_{\substack{\mathbf{m}: \forall j \in \bar{L} \cup \{i\}, m_j > 0 \\ \mathbf{m}_{L \setminus \{i\}} = \mathbf{0}_{|L \setminus \{i\}|}}} M_{\mathbf{x}}[\mathbf{m}] (\frac{-1}{|\bar{L} \cup \{i\}|}) \right).$$

Finally, we can pull $\frac{-1}{|\bar{L} \cup \{i\}|}$ out of the summation to be left with our final expression:

$$I_{\mathbf{x}}^{SV}(i) = - \sum_{\bar{L} \subseteq D \setminus \{i\}} \frac{1}{|\bar{L} \cup \{i\}| q^{|\bar{L} \cup \{i\}|}} \sum_{\substack{\mathbf{m}: \forall j \in \bar{L} \cup \{i\}, m_j > 0 \\ \mathbf{m}_{L \setminus \{i\}} = \mathbf{0}_{|L \setminus \{i\}|}}} M_{\mathbf{x}}[\mathbf{m}]. \quad (32)$$

This is equivalently written in vector notation as:

$$I_{\mathbf{x}}^{SV}(i) = - \sum_{\mathbf{k}: k_i > 0} \frac{1}{\|\mathbf{k}\|_0 q^{\|\mathbf{k}\|_0}} M_{\mathbf{x}}[\mathbf{k}]. \quad (33)$$

Analyzing the time complexity, in the worst-case scenario, if every $q$-ary Möbius coefficient is nonzero up to order $\ell$, the total amount of coefficients to sum up is $q^{\ell-1}(q - 1) \approx q^\ell$. Therefore, the time complexity of Equation (33) is upper-bounded by $\mathcal{O}(q^\ell)$.

### E.4 Proof of Theorem 3.5

The Faith-Shap equation given in Equation (2) is valid under the assumption that both $f$ and $I_{\mathbf{x}}^{FSI}(T)$ are constrained to a order $\ell$. If $f$ is not constrained by order $\ell$, we compute $I_{\mathbf{x}}^{FSI}(T)$ using the following equation [26]:

$$I^{FSI}(T) = a(v, T) + (-1)^{\ell-|T|} \frac{|T|}{\ell + |T|} \binom{\ell}{|T|} \times \sum_{\substack{S \supset T, \\ |S| > \ell}} \frac{\binom{|S|-1}{\ell}}{\binom{|S|+\ell-1}{\ell+|T|}} a(v, S), \quad \forall T \in T_\ell, \quad (34)$$

where $T_\ell$ denotes the set of all $T \subseteq D$ where $|T| \leq \ell$.

This formulation is not computationally scalable, as the set Möbius transform requires looping over $2^{|D|}$ subsets. However, for a $q$-ary Möbius transform with a maximum order $\ell$, all set Möbius

coefficients with a maximum order greater than $\ell$ are 0, by Equation (27). Therefore, when computing the $\ell^{\text{th}}$ order Faith-Shap interaction index, there are no set Möbius coefficients that satisfy $|S| > \ell$ in the summation term of Equation (34), allowing us to drop the summation term. Therefore, we can define the $\ell^{\text{th}}$ order Faith-Shap interaction index equation as follows:

$$I^{FSI}(T) = a(v, T), \quad \forall T \in T_\ell. \tag{35}$$

From Proposition 3.3, this is equivalently in vector notation as:

$$I^{FSI}(T) = (-1)^{|T|} \sum_{\mathbf{k}:\mathbf{k}_T > \mathbf{0}_{|T|}} \frac{1}{q^{\|\mathbf{k}\|_0}} M_{\mathbf{x}}[\mathbf{k}], \tag{36}$$

with a computational complexity of $\mathcal{O}(q^\ell)$.

Similarly to the computational complexity from Theorem 3.4, if every $q$-ary Möbius coefficient is nonzero up to order $\ell$, the total amount of coefficients to sum up is $q^{\ell - |T|}(q-1)^{|T|} \approx q^\ell$. Therefore, the time complexity of Theorem 3.5 is $\mathcal{O}(q^\ell)$. Notably, when $\ell = 1$, Theorem 3.5 becomes identical to computing SHAP values in Theorem 3.4.

# F    Additional Experimental Results

| Top positive SHAP values | | | Top negative SHAP values | | |
|---|---|---|---|---|---|
| **Position** | **Nucleotide** | **Average SHAP value** | **Position** | **Nucleotide** | **Average SHAP value** |
| 5 | G | 0.120 | 8 | A | -0.079 |
| 6 | G | 0.118 | 10 | A | -0.076 |
| 7 | G | 0.087 | 6 | C | -0.073 |
| 25 | T | 0.080 | 7 | A | -0.071 |
| 22 | T | 0.070 | 9 | A | -0.069 |
| 11 | C | 0.061 | 20 | C | -0.064 |
| 21 | T | 0.061 | 5 | T | -0.055 |
| 10 | C | 0.061 | 5 | C | -0.050 |
| 8 | G | 0.060 | 6 | T | -0.047 |
| 3 | C | 0.058 | 23 | C | -0.045 |
| 9 | G | 0.053 | 21 | C | -0.045 |
| 4 | C | 0.047 | 3 | T | -0.043 |
| 8 | C | 0.043 | 22 | A | -0.042 |
| 2 | A | 0.043 | 5 | A | -0.041 |
| 23 | T | 0.042 | 25 | C | -0.041 |
| 4 | G | 0.037 | 4 | T | -0.041 |
| 9 | C | 0.036 | 4 | A | -0.039 |
| 10 | G | 0.036 | 18 | C | -0.037 |
| 24 | G | 0.032 | 11 | A | -0.037 |
| 18 | A | 0.031 | 17 | C | -0.037 |

Table 2: Top 20 positive and negative SHAP zero values in TIGER from SHAP zero when predicting the guide score of perfect match target sequences. The top SHAP values are centered around the seed region and are in agreement with KernelSHAP.

| Top positive SHAP values | | | Top negative SHAP values | | |
|---|---|---|---|---|---|
| **Position** | **Nucleotide** | **Average SHAP value** | **Position** | **Nucleotide** | **Average SHAP value** |
| 6 | G | 0.123 | 10 | A | -0.066 |
| 5 | G | 0.119 | 7 | A | -0.065 |
| 7 | G | 0.094 | 8 | A | -0.062 |
| 11 | C | 0.076 | 9 | A | -0.062 |
| 25 | T | 0.073 | 20 | C | -0.060 |
| 4 | C | 0.070 | 6 | C | -0.058 |
| 10 | C | 0.062 | 3 | T | -0.046 |
| 8 | C | 0.059 | 6 | T | -0.044 |
| 3 | C | 0.055 | 9 | T | -0.041 |
| 8 | G | 0.054 | 3 | A | -0.040 |
| 9 | C | 0.052 | 11 | A | -0.040 |
| 21 | T | 0.050 | 5 | T | -0.040 |
| 22 | T | 0.049 | 8 | T | -0.038 |
| 7 | C | 0.049 | 4 | A | -0.038 |
| 9 | G | 0.047 | 22 | A | -0.037 |
| 4 | G | 0.046 | 23 | C | -0.037 |
| 12 | C | 0.039 | 21 | C | -0.037 |
| 18 | A | 0.038 | 10 | T | -0.036 |
| 10 | G | 0.037 | 5 | G | -0.036 |
| 23 | T | 0.036 | 5 | C | -0.035 |

Table 3: Top 20 positive and negative SHAP values in TIGER from KernelSHAP when predicting the guide score of perfect match target sequences. Similar to SHAP zero, the top SHAP values are centered around the seed region.

| Top positive interactions | | | Top negative interactions | | |
|---|---|---|---|---|---|
| **Positions** | **Nucleotides** | **Average interaction** | **Positions** | **Nucleotides** | **Average interaction** |
| 5,6 | G,T | 0.061 | 3,4 | C,C | -0.097 |
| 5,6 | C,A | 0.061 | 8,9 | C,C | -0.079 |
| 3,4 | A,C | 0.056 | 3,4 | G,G | -0.061 |
| 3,4 | C,G | 0.055 | 4,5 | G,G | -0.059 |
| 19,20 | C,A | 0.052 | 4,5 | C,C | -0.059 |
| 5,6 | A,C | 0.052 | 6,7 | C,C | -0.059 |
| 5,6 | T,G | 0.052 | 6,7 | G,G | -0.059 |
| 19,20 | A,C | 0.046 | 5,6 | G,G | -0.059 |
| 4,5 | T,G | 0.044 | 5,6 | C,C | -0.059 |
| 4,5 | A,C | 0.044 | 9,10 | T,T | -0.059 |
| 8,9 | A,C | 0.043 | 19,20 | A,A | -0.059 |
| 18,19 | C,A | 0.041 | 10,11 | G,G | -0.057 |
| 4,5 | G,T | 0.038 | 10,11 | C,C | -0.057 |
| 4,5 | C,A | 0.038 | 9,10 | C,C | -0.056 |
| 10,11 | A,C | 0.038 | 11,12 | G,G | -0.056 |
| 10,11 | T,G | 0.038 | 11,12 | C,C | -0.056 |
| 8,9 | C,G | 0.038 | 8,9 | G,G | -0.054 |
| 11,12 | A,C | 0.037 | 7,8 | T,T | -0.052 |
| 11,12 | T,G | 0.037 | 7,8 | G,G | -0.052 |
| 6,7 | T,G | 0.037 | 7,8 | C,C | -0.052 |

Table 4: Top 20 positive and negative Faith-Shap interactions in TIGER from SHAP zero when predicting the guide score of perfect match target sequences. The top interactions exhibit high GC content around the seed region.

| Top positive SHAP values | | | Top negative SHAP values | | |
|---|---|---|---|---|---|
| **Position** | **Nucleotide** | **Average SHAP value** | **Position** | **Nucleotide** | **Average SHAP value** |
| 19 | T | 3.727 | 19 | G | -5.462 |
| 19 | A | 2.466 | 18 | G | -3.105 |
| 20 | G | 2.431 | 20 | C | -2.981 |
| 21 | A | 1.833 | 21 | C | -2.891 |
| 22 | T | 1.754 | 18 | A | -2.617 |
| 22 | G | 1.751 | 21 | A | -2.539 |
| 18 | T | 1.734 | 17 | C | -2.495 |
| 21 | C | 1.733 | 17 | A | -2.495 |
| 17 | G | 1.696 | 17 | T | -2.461 |
| 20 | A | 1.649 | 22 | A | -2.451 |
| 18 | A | 1.527 | 23 | C | -2.266 |
| 17 | C | 1.492 | 21 | G | -2.239 |
| 21 | G | 1.490 | 22 | G | -2.223 |
| 18 | C | 1.490 | 18 | C | -2.138 |
| 21 | T | 1.422 | 22 | T | -2.120 |
| 22 | C | 1.355 | 22 | C | -2.034 |
| 23 | C | 1.220 | 20 | T | -2.001 |
| 16 | C | 1.177 | 17 | G | -1.972 |
| 22 | A | 1.121 | 20 | G | -1.877 |
| 17 | A | 1.104 | 18 | T | -1.775 |

Table 5: Top 20 positive and negative SHAP values in inDelphi from SHAP zero when predicting the frameshift frequency of DNA target sequences. Top SHAP values are centered around the cut site and are in agreement with KernelSHAP.

| Top positive SHAP values | | | Top negative SHAP values | | |
|---|---|---|---|---|---|
| **Position** | **Nucleotide** | **Average SHAP value** | **Position** | **Nucleotide** | **Average SHAP value** |
| 19 | T | 4.383 | 19 | G | -6.148 |
| 17 | G | 4.308 | 18 | G | -4.731 |
| 20 | G | 3.585 | 23 | C | -3.483 |
| 18 | T | 2.773 | 17 | C | -3.212 |
| 19 | A | 2.649 | 20 | C | -2.650 |
| 20 | A | 2.497 | 21 | C | -2.641 |
| 22 | G | 2.359 | 22 | T | -2.534 |
| 21 | T | 2.253 | 21 | A | -2.507 |
| 14 | G | 2.237 | 17 | T | -2.458 |
| 21 | A | 2.161 | 19 | C | -2.305 |
| 18 | C | 2.156 | 22 | A | -2.282 |
| 18 | A | 2.117 | 20 | G | -2.247 |
| 21 | C | 2.085 | 15 | G | -2.235 |
| 17 | C | 2.013 | 21 | G | -2.216 |
| 22 | T | 1.999 | 20 | T | -2.164 |
| 23 | C | 1.921 | 18 | A | -2.144 |
| 21 | G | 1.838 | 18 | C | -2.116 |
| 15 | T | 1.648 | 26 | G | -2.091 |
| 14 | T | 1.580 | 15 | T | -2.009 |
| 23 | G | 1.528 | 22 | G | -1.959 |

Table 6: Top 20 positive and negative SHAP values in inDelphi from KernelSHAP when predicting the frameshift frequency of DNA target sequences. Like SHAP zero, the top SHAP values are centered around the cut site.

| Top positive interactions | | | Top negative interactions | | |
|---|---|---|---|---|---|
| **Positions** | **Nucleotides** | **Average interaction** | **Positions** | **Nucleotides** | **Average interaction** |
| 18,19,21,22 | G,C,G,G | 2.504 | 18,19,21,22 | T,C,T,C | -5.640 |
| 18,19,21,22 | C,G,C,C | 2.504 | 18,19,21,22 | A,G,A,G | -5.640 |
| 18,19,21,22 | A,G,A,C | 2.504 | 18,19,21,22 | G,C,G,C | -5.640 |
| 18,19,21,22 | T,C,T,G | 2.504 | 18,19,21,22 | C,G,C,G | -5.640 |
| 18,21 | G,C | 2.073 | 18,19,21,22 | G,G,G,G | -5.640 |
| 18,21 | C,G | 2.073 | 18,21 | G,G | -5.325 |
| 17,18,21,22 | G,C,G,C | 1.983 | 18,21 | C,C | -5.325 |
| 17,18,21,22 | G,G,G,G | 1.983 | 17,18,20,21 | G,A,G,A | -4.658 |
| 17,18,21,22 | A,A,A,A | 1.983 | 17,18,20,21 | T,G,T,G | -4.658 |
| 17,18,21,22 | A,G,A,G | 1.983 | 17,18,20,21 | A,T,A,T | -4.658 |
| 17,18,21,22 | A,C,A,C | 1.983 | 17,18,20,21 | C,A,C,A | -4.658 |
| 17,18,21,22 | T,T,T,T | 1.983 | 17,18,20,21 | T,C,T,C | -4.658 |
| 17,18,21,22 | T,A,T,A | 1.983 | 17,18,20,21 | G,G,G,G | -4.658 |
| 18,19,20,21 | C,T,C,T | 1.982 | 17,18,20,21 | C,G,C,G | -4.658 |
| 18,19,20,21 | A,G,A,G | 1.982 | 17,18,20,21 | G,C,G,C | -4.658 |
| 18,19,20,21 | C,A,C,A | 1.982 | 17,18,20,21 | A,G,A,G | -4.658 |
| 18,19,20,21 | G,G,G,G | 1.982 | 17,18,20,21 | A,A,A,A | -4.658 |
| 18,19,20,21 | G,A,G,A | 1.982 | 17,18,20,21 | T,T,T,T | -4.658 |
| 18,19,20,21 | T,G,T,G | 1.982 | 17,18,20,21 | T,A,T,A | -4.658 |
| 18,19,20,21 | A,C,A,C | 1.982 | 18,21 | T,T | -4.553 |

Table 7: Top 20 positive and negative Faith-Shap interactions in inDelphi from SHAP zero when predicting the frameshift frequency of DNA target sequences. GCGC and TCTC interactions at positions 17-18 and 21-22 and at 18-19 and 21-22, respectively, show high-order microhomology patterns.

| Top positive SHAP values | | | Top negative SHAP values | | |
|---|---|---|---|---|---|
| **Position** | **Amino acid** | **Average SHAP value** | **Position** | **Amino acid** | **Average SHAP value** |
| 8 | L | 0.074 | 8 | D | -0.046 |
| 4 | L | 0.074 | 4 | E | -0.044 |
| 4 | I | 0.060 | 4 | D | -0.044 |
| 6 | C | 0.058 | 5 | P | -0.041 |
| 5 | F | 0.058 | 4 | P | -0.040 |
| 8 | V | 0.053 | 4 | K | -0.039 |
| 8 | I | 0.050 | 4 | N | -0.039 |
| 8 | M | 0.050 | 4 | R | -0.037 |
| 4 | M | 0.049 | 8 | E | -0.036 |
| 2 | F | 0.046 | 8 | G | -0.036 |
| 5 | I | 0.042 | 9 | P | -0.036 |
| 0 | F | 0.040 | 7 | W | -0.035 |
| 5 | Y | 0.037 | 9 | W | -0.035 |
| 7 | V | 0.037 | 5 | D | -0.034 |
| 1 | M | 0.035 | 8 | P | -0.034 |
| 3 | K | 0.034 | 4 | Q | -0.033 |
| 4 | V | 0.034 | 6 | W | -0.032 |
| 4 | A | 0.032 | 8 | N | -0.031 |
| 9 | K | 0.032 | 8 | K | -0.031 |
| 1 | F | 0.032 | 5 | E | -0.030 |

Table 8: Top 20 positive and negative SHAP values from SHAP zero when predicting protein fitness of protein sequences in Tranception. Top SHAP values are consistent with secondary structure predictors and are in agreement with KernelSHAP.

| Top positive SHAP values | | | Top negative SHAP values | | |
|---|---|---|---|---|---|
| **Position** | **Amino acid** | **Average SHAP value** | **Position** | **Amino acid** | **Average SHAP value** |
| 5 | F | 0.012 | 8 | D | -0.108 |
| 4 | L | 0.008 | 4 | D | -0.107 |
| 6 | C | 0.007 | 4 | E | -0.103 |
| 8 | L | 0.006 | 8 | E | -0.101 |
| 1 | M | 0.006 | 4 | P | -0.100 |
| 2 | F | 0.005 | 4 | N | -0.100 |
| 0 | F | 0.005 | 8 | P | -0.099 |
| 9 | K | 0.005 | 4 | K | -0.096 |
| 3 | K | 0.003 | 5 | P | -0.095 |
| 7 | V | 0.003 | 4 | R | -0.095 |
| - | - | - | 8 | G | -0.094 |
| - | - | - | 8 | K | -0.093 |
| - | - | - | 8 | N | -0.093 |
| - | - | - | 5 | D | -0.092 |
| - | - | - | 8 | W | -0.090 |
| - | - | - | 4 | Q | -0.090 |
| - | - | - | 5 | E | -0.089 |
| - | - | - | 6 | W | -0.082 |
| - | - | - | 8 | R | -0.080 |
| - | - | - | 4 | G | -0.080 |

Table 9: Top positive and negative SHAP values from KernelSHAP in Tranception. KernelSHAP is biased toward negative SHAP values due to marginalization in training data.

| Top positive interactions | | | Top negative interactions | | |
|---|---|---|---|---|---|
| **Positions** | **Amino acids** | **Average interaction** | **Positions** | **Amino acids** | **Average interaction** |
| 4,6 | R,Y | 0.013 | 4,6 | K,E | -0.017 |
| 4,6 | K,W | 0.011 | 2,3 | F,W | -0.011 |
| 5,9 | K,P | 0.011 | 4,6 | G,F | -0.011 |
| 5,9 | D,W | 0.011 | 5,9 | K,K | -0.011 |
| 2,3 | F,K | 0.010 | 7,9 | V,P | -0.010 |
| 5,7 | P,N | 0.009 | 5,9 | W,I | -0.010 |
| 4,6 | I,E | 0.009 | 4,6 | K,Q | -0.008 |
| 4,6 | K,Y | 0.008 | 5,7 | K,L | -0.008 |
| 5,9 | F,K | 0.008 | 4,6 | Y,T | -0.008 |
| 2,3 | F,R | 0.008 | 2,3 | F,F | -0.008 |
| 5,7 | W,G | 0.007 | 2,3 | F,I | -0.008 |
| 4,6 | K,F | 0.007 | 5,9 | E,Q | -0.008 |
| 2,3 | F,Q | 0.007 | 5,7 | K,I | -0.008 |
| 4,6 | N,W | 0.007 | 5,7 | K,M | -0.008 |
| 4,6 | G,H | 0.007 | 5,7 | A,N | -0.007 |
| 5,7 | K,E | 0.006 | 5,7 | P,F | -0.007 |
| 7,9 | N,P | 0.006 | 5,9 | K,D | -0.007 |
| 5,7 | K,Q | 0.006 | 5,7 | K,V | -0.007 |
| 5,9 | K,F | 0.006 | 4,6 | W,T | -0.007 |
| 7,9 | P,P | 0.006 | 5,7 | G,C | -0.007 |

Table 10: Top 20 positive and negative Faith-Shap interactions from SHAP zero when predicting protein fitness in Tranception reveals epistatic interactions in secondary structure predictors.

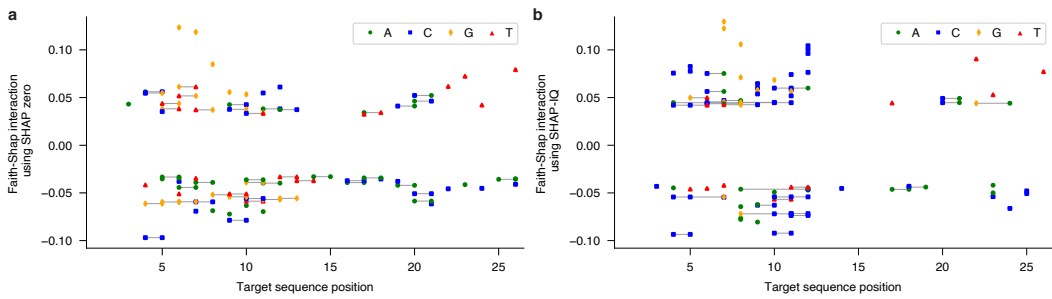

Figure 5: **Top interactions in TIGER. a**, Top 80 Faith-Shap interactions in TIGER with SHAP zero and **b**, SHAP-IQ. Although overlapping interactions in SHAP zero and SHAP-IQ are in agreement, SHAP zero interactions are more concentrated around the seed region.

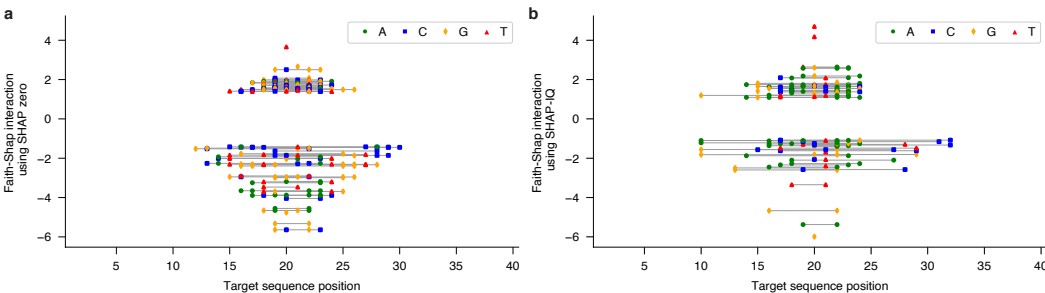

Figure 6: **Top interactions in inDelphi. a**, Top 80 Faith-Shap interactions in inDelphi with SHAP zero and **b**, SHAP-IQ. While overlapping interactions are in agreement, the majority of SHAP-IQ interactions are not centered around the cut site.

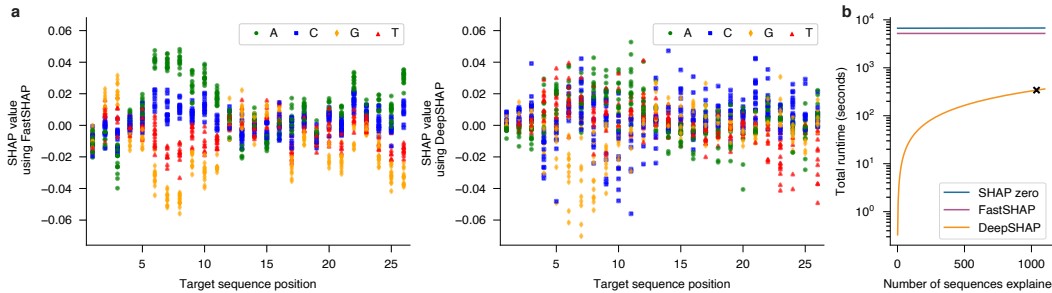

Figure 7: **FastSHAP and DeepSHAP on TIGER. a**, FastSHAP and DeepSHAP estimates over 1038 query sequences, with a random subset of 50 estimates shown for clarity. **b**, Total runtime of SHAP zero, FastSHAP, and DeepSHAP over 1038 target sequences. While both methods are faster than SHAP zero, FastSHAP estimates do not correlate well with KernelSHAP, and DeepSHAP estimates are not black-box and do not support amortized inference.

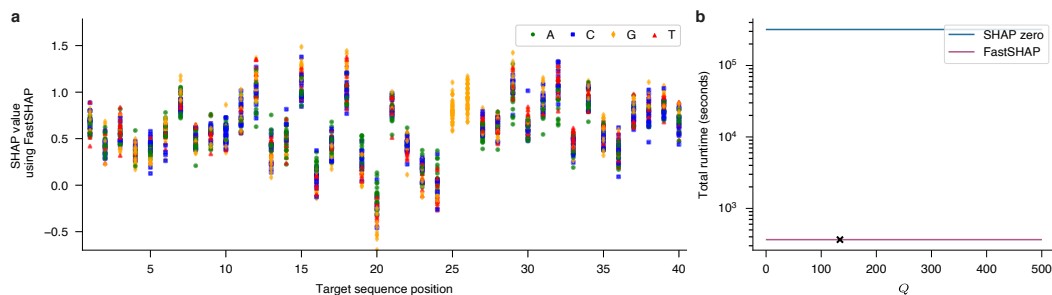

Figure 8: **FastSHAP on inDelphi. a**, SHAP values of 40 random target sequences from FastSHAP. **b**, Total runtime of SHAP zero and FastSHAP over 134 target sequences. While FastSHAP is faster than SHAP zero, FastSHAP estimates do not correlate well with KernelSHAP.

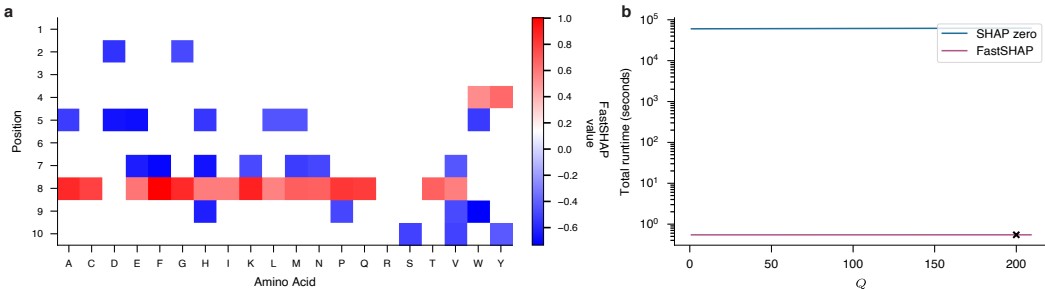

Figure 9: **FastSHAP on Tranception. a**, SHAP values on held-out sequences from FastSHAP. **b**, Total runtime of SHAP zero and FastSHAP over 200 sequences. While FastSHAP is faster than SHAP zero, FastSHAP estimates do not correlate well with KernelSHAP.

### F.1 Verifying Epistatic Interactions in Tranception

We evaluated whether the top five positive and negative interactions identified in Table 10 correspond to epistasis in Tranception. As computing ground-truth Faith-Shap interactions is too computationally expensive at the scales we run SHAP zero, we opted to use SHAP zero to predict the top mutations in sequences that would result in epistasis.

From the held-out sequence set (Appendix G.8), we first selected sequences containing the relevant amino acid pairs. For each pair, we use SHAP zero's Möbius transform to predict whether mutations to the amino acids pairs will results in epistasis, where second-order Möbius coefficients quantify pairwise interactions. We extracted the top three mutation pairs per interaction and introduced these mutations into the selected sequences.

To measure epistasis, we compared the Tranception scores of the original selected sequence $\mathbf{x}^0$, the corresponding single mutants $\mathbf{x}_i^1$ and $\mathbf{x}_j^1$ at positions $i$ and $j$, respectively, and the double mutant $\mathbf{x}_{i,j}^2$. Epistasis values $\epsilon$ were computed as [60]:

$$\epsilon = f(\mathbf{x}_{i,j}^2) - f(\mathbf{x}_i^1) - f(\mathbf{x}_j^1) + f(\mathbf{x}^0),$$

where $f(\cdot)$ denotes the Tranception score. By construction, $\epsilon$ is equivalent to the ground-truth second-order Möbius coefficient for the corresponding amino acid pair.

Table 11 reports the average epistasis value found in Tranception for these interactions. Here, we see that SHAP zero correctly identifies interactions where Tranception exhibits positive and negative epistasis.

| Top positive interactions | | | Top negative interactions | | |
|---|---|---|---|---|---|
| Positions | Amino acids | Epistasis | Positions | Amino acids | Epistasis |
| 4,6 | R,Y | 0.04 | 4,6 | K,E | -0.04 |
| 4,6 | K,W | 0.03 | 2,3 | F,W | -0.01 |
| 5,9 | K,P | 0.04 | 4,6 | G,F | -0.02 |
| 5,9 | D,W | 0.04 | 5,9 | K,K | -0.05 |
| 2,3 | F,K | 0.02 | 7,9 | V,P | -0.02 |

Table 11: Top positive and negative pairwise interactions identified by SHAP zero, with epistasis values from Tranception.

## G   Experimental Setup

Experiments that involve querying TIGER or Tranception were run on a single NVIDIA RTX A6000 machine. All other experiments were run on CPU.

### G.1   Implementation of Computing the Sparse Fourier Transform

In all experiments to recover Fourier coefficients, we applied the sparse Fourier algorithm $q$-SFT [40] through their GitHub repository (https://github.com/basics-lab/qsft). We specify which $q$ and $n$ we run in each experimental subsection. There are three tunable parameters in SHAP zero that determine the number of samples used: subsampling dimension $b$, number of subsampling matrices ($C$), and number of offset matrices ($P_1$). Given a length-$n$ sequence, these parameters determine the number of samples SHAP zero takes to recover the most significant Fourier coefficients: $samples = q^b \times C \times P_1 \times (n+1)$. Increasing $b$ increases the number of samples SHAP zero takes exponentially. The theory of compressed sensing tells us that the quality of recovered top Fourier coefficients depends on the total number of samples queried, meaning that performance improves monotonically with respect to all parameters. We also observe this in our empirical results: increasing $b$ results in recovering more Fourier coefficients and better SHAP zero accuracy. Therefore, to determine the number of samples to take in SHAP zero, we maximize $b$ as large as computationally feasible while leaving $C = 3$ and $P_1 = 3$ (which follows the default parameters of $q$-SFT), ensuring the recovery of the top Fourier coefficients. All other parameters in $q$-SFT were left as default,

which includes setting $query\_method$ to "complex", $delays\_method\_source$ to "identity", and $delays\_method\_channel$ to "nso".

After determining the number of samples to use, we then optimized the hyperparameter $noise\_sd$, which sets the peeling threshold to recover Fourier coefficients. To find the optimal value of $noise\_sd$, we performed a coarse search from 0 to 20 at 20 evenly spaced points and used the recovered Fourier coefficients at each level of $noise\_sd$ to predict the values of a test set of 10,000 samples (done by setting the test argument $n\_samples$ to $10,000$). After finding the general range where $R^2$ was the largest, we refined our search within that general range at a smaller step size. We report the specific hyperparameters used for $q$-SFT in each experimental subsection.

## G.2  Comparison With Baselines

Computing exact SHAP values or interactions for $q$-ary functions requires $\mathcal{O}(q^n)$ computations (see Appendix E.3 and E.4). For realistic values of $q$ and $n$ in sequence models, even a single evaluation is intractable; for example, estimating one SHAP value for inDelphi would take on the order of $10^{10}$ years on our servers. As a result, we benchmark SHAP zero against KernelSHAP and SHAP-IQ, two methods that yield unbiased Shapley estimations [11, 25, 62, 65]. To ensure fair comparisons, we conduct a thorough convergence analysis for both KernelSHAP and SHAP-IQ (see below), and verify that that both baselines produce stable explanations. In addition, we performed a detailed analysis of SHAP zero's explanations and uncovered interactions that align with known biological motifs from the literature. Together, these results demonstrate that SHAP zero produces biologically plausible and scalable amortized explanations for sequence models.

## G.3  Implementation of KernelSHAP, SHAP-IQ, and FastSHAP

To create the KernelSHAP baseline, we used the Python package SHAP's KernelSHAP implementation. For the background set, we used training sequences from inDelphi and TIGER. In Tranception, we used deep mutational scanning GFP data [49] from ProteinGym [66], and filtered out sequences that were mutated outside of the 10 sites chosen. In Tranception, we were able to run SHAP on the full background dataset, due to the manageable number of sequences present in the dataset. However, in the instance of TIGER and inDelphi, we were unable to feasibly run SHAP with all training sequences. Therefore, to determine the size of the background dataset, we ran KernelSHAP over different background dataset sizes on a subset of the held-out sequences and plotted the $R^2$ of the KernelSHAP estimates with respect to the largest background dataset size. Error bars are generated by computing the $R^2$ over each sequence's KernelSHAP estimates. We picked the background dataset size where KernelSHAP values converged with an $R^2 > 0.97$ to run on all the held-out sequences.

To compute Faith-Shap interactions, we used SHAP-IQ's implementation, available on their GitHub repository (https://github.com/mmschlk/shapiq). Since SHAP zero's recovered interactions were greater than 3 in both TIGER and inDelphi, we wanted to run SHAP-IQ at $max\_order$ $(\ell) > 3$. However, we ran into computational issues in both cases, resulting in us setting $\ell = 3$. In Tranception, we set $\ell = 2$, since SHAP zero wasn't unable to recover significant interactions greater than 2. Additionally, since SHAP-IQ was computationally too expensive to run over all sequences, we randomly selected a few sequences to analyze. We used the same background dataset from KernelSHAP and hyperparameter tuned $budget$ (which determines the number of model evaluations to make). Similar to KernelSHAP, we ran SHAP-IQ over different budgets on a subset of held-out sequences and plotted the $R^2$ of the SHAP-IQ interactions with respect to the largest budget size. Error bars are generated by computing the $R^2$ over each sequence's Faith-Shap estimates, stratified by interaction order. We picked the budget where Faith-Shap estimates converged with an $R^2 > 0.6$ to run on all the held-out sequences. In the case of Tranception, we noticed that the Faith-Shap interactions converged quickly to $R^2 = 1$ at $budget = 10,000$. Therefore, we opted to use $budget = 10,000$ for our experiments. We report the specific KernelSHAP and Faith-Shap parameters for TIGER and inDelphi in each experimental subsection. Fig. 10 shows the convergence plots for all three models, with dashed lines showing what experimental parameters were used.

To run FastSHAP, we used the FastSHAP implementation available on their GitHub repository (https://github.com/iancovert/fastshap). We removed all instances of calling the $softmax$ function. We replaced the KL divergence loss used to train the surrogate neural network for mean squared error loss since the FastSHAP implementation on their GitHub is designed for classification and not

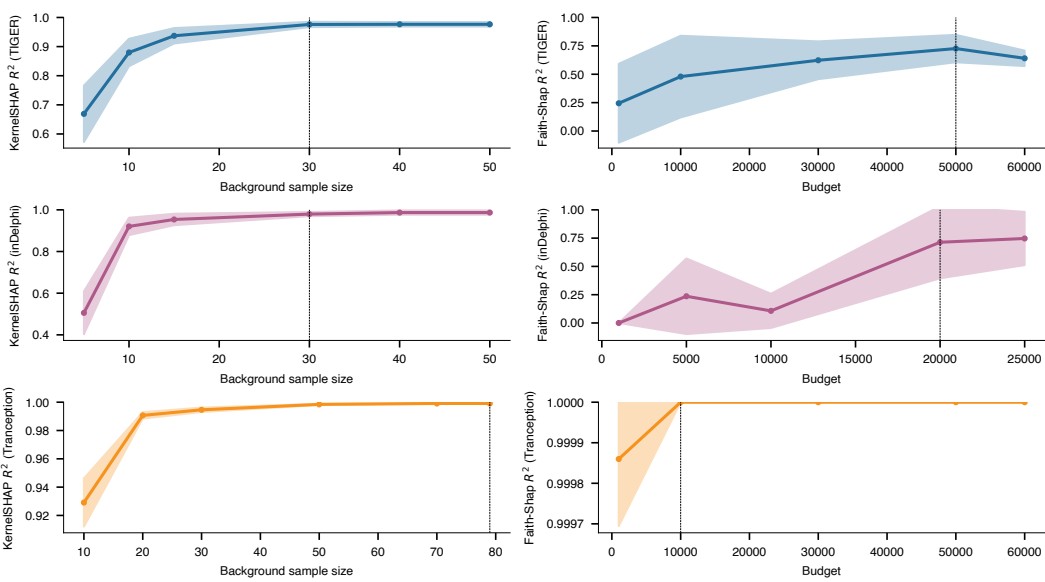

Figure 10: **Shapley convergence plots in TIGER, inDelphi, and Tranception.** We hyperparameter tuned the size of the background dataset in KernelSHAP and *budget* in SHAP-IQ. Dashed lines indicate what experimental parameters were used.

regression. Following the FastSHAP implementation left on their GitHub for the census dataset, the surrogate model is implemented using a neural network consisting of three fully connected layers, where each layer is 128 units wide, with ReLU activation. We did not train the surrogate model according to their CIFAR implementation because their CIFAR-10 surrogate model uses a pre-trained ResNet-50 model designed for image-specific tasks. The surrogate model was trained on the entire training set. All other FastSHAP parameters were left as default according to their GitHub implementation on the CIFAR-10 dataset.

### G.4    Calculation of Top SHAP Values, Top Faith-Shap Interactions, and Total Runtime

To determine the top SHAP values in the Appendix (see Tables 2, 3, 5, 6, 8, and 9), we separated the SHAP values into positive and negative values. For the positive and negative grouping, we calculated the average contribution of each nucleotide at each position to report the average SHAP value. The same procedure was used to determine the top Faith-Shap interactions (see Tables 4, 7, and 10), but instead by calculating the average contribution of each unique interaction. Total runtime was computed by summing the sample and computational complexities of each algorithm. We estimated the runtime per sample by dividing the total runtime by the number of query explanations performed.

### G.5    Visualization of SHAP Values and Faith-Shap Interactions in TIGER and inDelphi

To visualize SHAP values (see Fig. 2c, Fig. 3c, Fig. 7a, and Fig. 8a), we randomly sampled 50 sequences and plotted their SHAP values on a scatterplot. To plot a histogram of Faith-Shap interactions (see Fig. 2e and 3e), we used the visualization scheme described in [28]. After separating the positive and negative interactions, we evenly distributed the interaction magnitudes across all affected nucleotides. We reported the summed contributions of each nucleotide at each position on a bar graph. However, slightly deviating from [28], rather than stratifying the bar plots by the order of interactions, we stratified them by nucleotide type and only plot interactions with an order greater than 1. For more details, refer to Appendix B of [28]. To visualize Faith-Shap interactions as shown in the Appendix (see Fig. 5 and 6), we plotted interactions as data points with line connecting points to show the order of interaction. For visual clarity, we only plotted the larger interaction if interactions with the same nucleotides at the same positions have an estimated difference of 0.001. We plotted the top 80 interactions from SHAP zero and SHAP-IQ.

### G.6 TIGER

We considered the sequence-only implementation of TIGER by inputting length-$n = 26$ perfect match target sequences, where the model assumes that the gRNA is a perfect sequence complement of the target sequence. We set $q = 4$, $n = 26$, and $b = 7$. For $noise\_sd$, after finding the general range where $R^2$ was the largest, we refined our search using a step size of 0.025. We used $b = 7$ and $noise\_sd = 0.725$, which corresponded to an $R^2$ of 0.55.

We used the held-out perfect match target sequences provided by TIGER [33] to predict the experimental scores. We evaluated the performance of the recovered Fourier coefficients using Pearson correlation, Spearman correlation, area under the receiver operating characteristic curve (AUROC), and area under the precision-recall curve (AUPRC). AUROC and AUPRC were used because TIGER allows thresholding predictions to formulate a binary classification problem. We created our linear and pairwise interaction model baselines using sklearn in Python. Our baseline models were trained on one-hot encoded sequences using the training data from TIGER to predict the experimental guide score, employing Ridge regression with $\lambda = 0.5$. To construct the pairwise interaction model, we used sklearn's PolynomialFeatures implementation, setting $degree$ to 2 and $interaction\_only$ to True.

For KernelSHAP and SHAP-IQ, background samples were randomly drawn from the training set. During hyperparameter tuning, we evaluated KernelSHAP using 5, 10, 15, 30, 40, and 50 background samples across 30 held-out sequences; we selected 30 background samples for our experiments. For SHAP-IQ, we performed a hyperparameter search over budgets of 1,000, 10,000, 30,000, 50,000, 60,000, and 70,000 on two held-out sequences; we chose a budget of 50,000 for downstream analysis. Due to SHAP-IQ's high computational cost, we limited evaluation to a randomly selected subset of eight sequences. We additionally set $max\_order = 3$: the highest order feasible before encountering computational constraints. SHAP zero was run on all held-out sequences.

DeepSHAP was implemented using the Python package SHAP. The background dataset consists of 5000 samples randomly drawn from the training set, which is the number of background samples TIGER utilizes in their own DeepSHAP implementation. All other parameters were left as default.

### G.7 inDelphi

We applied SHAP zero on inDelphi by inputting $n = 40$ DNA target sequences over a $q = 4$ alphabet, which were additionally padded with 20 nucleotides on each end due to inDelphi's minimum length requirement, to get their frameshift frequency. We refined our $noise\_sd$ estimate using a step size of 0.1. We used $b = 7$ and $noise\_sd = 15$, which corresponded to an $R^2$ of 0.82.

We used the held-out DNA target sequences provided by inDelphi [34] to predict the experimental scores. We evaluated the performance of the recovered Fourier coefficients using Pearson and Spearman correlation. Since inDelphi does not have an option to threshold predictions like in TIGER, we do not report AUROC or AUPRC. The baseline models were trained on one-hot encoded sequences using the training data from inDelphi to predict inDelphi's predicted frameshift frequency, as the experimental frameshift frequencies were not provided. KernelSHAP, SHAP-IQ, and FastSHAP were created using the same procedure as in the TIGER experiment.

SHAP zero was run on all held-out sequences plus an additional 50 training sequences due to the limited amounts of held-out data to compute SHAP values and Faith-Shap interactions. For hyperparameter tuning, we evaluated KernelSHAP using 5, 10, 15, 30, 40, and 50 background samples across 10 held-out sequences, and selected 30 background samples for our experiments. To hyperparameter tune SHAP-IQ, we performed a hyperparameter search over budgets of 1,000, 5,000, 10,000, 20,000, 25,000, 30,000 on one held-out sequence, and chose a budget of 20,000 for our experiments. We limited our final SHAP-IQ experiments to run on a randomly selected subset of four sequences. Similar to TIGER, we additionally set $max\_order = 3$ due to computational constraints.

### G.8 Tranception

Unlike TIGER and inDelphi, Tranception takes in protein sequences. We consider the "large" model on Tranception with retrieval, which has 700 million parameters [35]. Tranception with retrieval scores each mutant by taking a weighted average of two values: the log-likelihood from the main transformer model and the log-likelihood derived from the multiple sequence aligment (MSA) at the

mutant's position, computed from the empirical amino acid distribution. When applying SHAP zero on Tranception, the MSA is initialized once during the beginning and remains fixed throughout the duration of sampling, ensuring a consistent sequence-function model.

We feed in the cannonical wildtype GFP sequence (which can be found at [66]) and mutate the $n = 10$ epistatic sites [49] experimentally tested (located in Supplementary Table 1). We additionally use the MSA provided from ProteinGym [66] for retrieval. We refine our $noise\_sd$ estimate using a step size of $0.01$. We used $b = 3$ and $noise\_sd = 0.03$, which corresponded to an $R^2$ of $0.97$.

SHAP zero was evaluated on 200 held-out sequences, randomly generated by introducing up to five mutations from the wildtype. To ensure comprehensive coverage, each amino acid appeared at least once at every position across the held-out set. We opted for synthetic test sequences because the deep mutational scanning (DMS) dataset from [49] lacked full coverage of all amino acids across all positions, which would have limited the effectiveness of SHAP zero to find epistatic interactions. KernelSHAP and SHAP-IQ are run using the DMS dataset from [49], containing 79 sequences, over the sites chosen. Since $n = 10$, we were able to run KernelSHAP and SHAP-IQ over all of the DMS data without computational issues. We run SHAP-IQ with $max\_order = 2$ and $budget = 10,000$ over 50 held-out sequences.

To create the linear and pairwise baselines, our baseline models were trained on our 200 sequence dataset to predict Tranception's score. We opted to use these sequences specifically, as Tranception was trained in an self-supervised fashion, and does not have any explicit sequence-function training data unlike TIGER and inDelphi. We then tested SHAP zero, linear, and pairwise models to predict experimental protein function from the DMS sequences.

### G.9 Visualization of SHAP Values and Faith-Shap Interactions in Tranception

To visualize SHAP values (see Fig. 4c and Fig. 9a), we computed the average SHAP value per amino acid at each position, and plotted the top 20% of the positive and negative SHAP values on a heatmap. In the case of KernelSHAP, we plotted all the positive SHAP values on the heatmap, since we observed that KernelSHAP was biased toward negative values (see Discussion). To visualize Faith-Shap interactions (see Fig. 4e), we computed the average pairwise interaction of amino acids. We plotted the top 20% of the positive and negative interactions on the heatmap.

