# OpenReview forum: "SHAP zero Explains Biological Sequence Models with Near-zero Marginal Cost for Future Queries"
_NeurIPS.cc/2025/Conference — NeurIPS 2025 poster_

### Official Review · Reviewer_GnFL · 2025-07-01

**Clarity:** 1
**Significance:** 1
**Originality:** 2
**Rating:** 4
**Confidence:** 3

**Summary:**

In this paper, the authors introduce SHAP zero, a fast approximate method for computing SHAP values for model explainability based on sparse Fourier transforms and near-zero marginal cost for queries over large biological sequence datasets. The method is evaluated on three sequence-based models for predictive tasks related to guide RNA efficacy, DNA repair, and protein fitness. SHAP zero shows comparable results to KernelSHAP, while being many times faster.

**Questions:**

The manuscript repeatedly refers to a model “sketch”, and specifies that the Fourier transform of a black-box model f is the “global sketch” of that model. What is the exact definition and meaning of “sketch” in this context?
What is the rationale for excluding gradient-based attribution methods, considering that all of the models f in the work are differentiable neural networks?

**Ethical Concerns:**

["NO or VERY MINOR ethics concerns only"]

**Final Justification:**

The authors' response has addressed my comments and questions and I will increase my score accordingly.

**Limitations:**

Yes

**Quality:**

2

**Strengths And Weaknesses:**

Strengths
The method is mainly compared against KernelSHAP, and seems to give a significant speedup with comparable results for sequence-based models.
Model explainability is an important and understudied problem for biological sequence models.

Weaknesses
The notation for Shapley values is appreciated, but probably better suited to a Methods or Preliminaries section than almost immediately in the Introduction.
The main text would benefit from pseudocode showing the SHAP zero algorithm.
Overall, the connections between the sparse Fourier transform, Mobius transform, and SHAP values are mathematically clear, but not intuitively clear. In addition to writing out the algorithm, it may be beneficial to simply explain how traditional SHAP and SHAP zero differ.
There is no mention of gradient-based attribution methods, like integrated gradients, which are possibly the most widely adopted and easy to use methods for explaining sequence-based models.

---

> ### Author Rebuttal · Authors · 2025-07-29
>
> We thank reviewer GnFL for their comments. We address the following questions:
>
> > The notation for Shapley values is appreciated, but probably better suited to a Methods or Preliminaries section than almost immediately in the Introduction. The main text would benefit from pseudocode showing the SHAP zero algorithm.
>
> We will provide a more thorough explanation of Shapley values and add a SHAP zero algorithm box to the main text.
>
> > Overall, the connections between the sparse Fourier transform, Mobius transform, and SHAP values are mathematically clear, but not intuitively clear.
>
> Briefly, the connection between SHAP values, the Möbius transform, and the sparse Fourier transform lies in how we represent and recover feature interactions in complex models. SHAP values attribute a model’s prediction to individual features and their combinations by decomposing the function into additive contributions over subsets. This decomposition aligns with the Möbius transform, a mathematical framework that captures how a function defined over feature subsets can be expressed in terms of unique contributions from each subset. However, directly computing this is intractable for high-dimensional models. The sparse Fourier transform offers a computational shortcut by exploiting the empirical sparsity of interactions in biological systems—only a few feature combinations significantly affect the outcome.
>
> We have made this connection clear throughout the paper, specifically around lines 52, 153, and 182. To address the reviewer's concern, we will add additional explanations to make the connection more clear.
>
> > The manuscript repeatedly refers to a model “sketch”, and specifies that the Fourier transform of a black-box model $f$ is the “global sketch” of that model. What is the exact definition and meaning of “sketch” in this context?
>
> We formally defined the sketch as the $s$-sparse approximation of the Fourier transform of the model, as detailed in Equation (3). We refer to this as a "global" sketch because this Fourier representation describes the entire function's behavior, independent of any single input query.
>
> We clarify that the term "sketch" is standard terminology in theoretical computer science and machine learning. The core idea of a sketch is to create a compressed summary of a model that captures the most relevant information. We are happy to make this more clear in the text.
>
> > What is the rationale for excluding gradient-based attribution methods, considering that all of the models $f$ in the work are differentiable neural networks?
>
> We clarify that the focus of our work is on amortizing Shapley-based explanations in **black-box sequence models**. Gradient-based attribution methods, which rely on model internals, fall under white-box explainability methods. Regardless, we have addressed white-box explainability methods in the Discussion section, and additionally provided experiments in the Appendix on running DeepSHAP, another white-box method, on TIGER in Appendix D.
>
> We emphasize that with the growth of large-scale AI sequence models with proprietary access, the choice of black-box explainability methods over white-box ones is more of a need than a luxury for biological discoveries.

---

> > ### Comment · Reviewer_GnFL · 2025-08-05
> >
> > The authors' response has addressed my comments and questions and I will increase my score accordingly.

---

### Official Review · Reviewer_b3je · 2025-07-01

**Clarity:** 3
**Significance:** 3
**Originality:** 3
**Rating:** 5
**Confidence:** 2

**Summary:**

This paper introduces SHAP zero, an algorithm for explaining black-box biological sequence models. By amortizing the computational cost of calculating Shapley values across many queries, the method enables large-scale interpretability studies that were previously intractable. The main idea is to create a one-time global sketch of the model using a sparse Fourier transform, from which local explanations for future queries can be derived at near-zero marginal cost. The authors demonstrate its effectiveness on models for gRNA efficacy, DNA repair, and protein fitness.

**Questions:**

- In the analysis of the Tranception model, the paper notes the use of "Tranception with retrieval". This model's function is not a simple mapping from a sequence X to a real value R, i.e., f:X→R. Instead, it incorporates information from a MAS, making the function closer to f:(X,M)→R, where M represents the MSA. Could the authors clarify how this additional input from the MSA space was handled within the SHAP zero framework?

- The validation is focused on DNA and protein sequence models. Could the authors comment on the potential applicability of SHAP zero to other types of biological data that also have a combinatorial nature but a different structure, such as protein-protein interaction networks or models of RNA-protein binding?

**Ethical Concerns:**

["NO or VERY MINOR ethics concerns only"]

**Final Justification:**

My concerns are addressed, and I will be glad to see the XAI in the area of AI4Sci.

**Limitations:**

- The authors have provided a thoughtful discussion of the limitations of their work, particularly regarding the assumptions of Fourier sparsity and the uniform marginalization strategy. I find this discussion to be transparent and appropriate.

**Paper Formatting Concerns:**

No concerns.

**Quality:**

3

**Strengths And Weaknesses:**

Strengths
- The manuscript is well-written and presents an exciting application of model interpretability to the important domain of biological sequences. The connection established between Shapley values, the Möbius transform, and the sparse Fourier transform is elegant and provides a strong theoretical foundation for the proposed method.
- The case study involving the Tranception model is interesting. The method demonstrates a promising capability to uncover the mechanistic basis of epistasis in Green Fluorescent Protein (GFP), a task of significant biological interest.

Weaknesses
- While the results are promising, the claims about capturing epistatic interactions would be substantially strengthened by including a broader range of examples. The paper presents a single case study on GFP using the Tranception model. To more robustly validate the method's ability to uncover epistasis, it would be beneficial to apply it to other proteins with well-characterized or computationally predicted epistatic networks
- A fundamental challenge is the evaluation of the generated explanations. The manuscript demonstrates that the explanations align with some known biological phenomena, but the concept of a "ground truth" for interpretability remains elusive. How can we quantitatively assess the correctness of an explanation, especially in cases where the underlying biological mechanisms are not fully understood by human experts? Without a clear evaluation framework, the interpretations risk being perceived as plausible yet speculative hypotheses.

---

> ### Author Rebuttal · Authors · 2025-07-29
>
> We thank reviewer b3je for taking the time to review our paper. We address the following questions:
>
> > Could the authors clarify how this additional input from the MSA space was handled within the SHAP zero framework (for Tranception with retrieval)?
>
> In our framework, the multiple sequence alignment (MSA) is treated as a fixed component of the overall black-box model, not as a separate input for each query.
>
> Tranception with retrieval scores each mutant by taking a weighted average of two values: the log-likelihood from the main transformer model and the log-likelihood derived from the MSA at the specific mutant's position (see Fig. 3 and Equation (4) in [1]). The MSA-derived likelihood is based on the empirical amino acid distribution at that position.
>
> When using SHAP zero, Tranception with retrieval is treated as a single, consistent black-box function. The MSA is initialized only once at the beginning and remains static for all subsequent mutant evaluations. The user does not need to re-initialize or change the MSA for every new mutant, ensuring our definition of the sequence-function model from the Introduction remains consistent.
> We will add a summary of this discussion in the Appendix.
>
> > The validation is focused on DNA and protein sequence models. Could the authors comment on the potential applicability of SHAP zero to other types of biological data that also have a combinatorial nature but a different structure, such as protein-protein interaction networks or models of RNA-protein binding?
>
> Thank you for your question. SHAP zero can be applied to other combinatorial biological problems. For problems such as protein-protein binding, the application is seamless: we would frame the input as the combined sequence of interacting proteins and the output as the binding affinity. Assuming this mapping is sparse in Fourier (i.e., a small fraction of residue-residue interactions contribute to binding), SHAP zero will generalize. For problems such as RNA-protein binding, an interesting extension of SHAP zero would be to compute the Fourier transform over different alphabet sizes ($q$=4 and $q$=20) using generalized $q$-ary Fourier algorithms [2].
>
> [1] Notin et al. “Tranception: protein fitness prediction with autoregressive transformers and inference-time retrieval” (2022)
>
> [2] Tsui et al. “Efficient Algorithm for Sparse Fourier Transform of Generalized $q$-ary Functions” (2025)

---

> > ### Comment · Reviewer_b3je · 2025-08-04
> >
> > Most of my concerns are addressed and I will keep my score.

---

### Official Review · Reviewer_hhqE · 2025-07-02

**Clarity:** 4
**Significance:** 3
**Originality:** 3
**Rating:** 5
**Confidence:** 4

**Summary:**

The authors describe a method for computing Shapley values for biological sequence-to-function models with query-only access that scales significantly better than existing exponential-time methods. It achieves this by paying an upfront polynomial-time cost to create a global Fourier sketch of the model, which is tractable for biological sequence contexts given their sparsity. The authors then show that for a given sequence, the Fourier coefficients can be mapped to Shapley values via a Mobius transform that scales with O(q^l), where q is the dimensionality of the input space (e.g. 4 for nucleic acids and 20 for proteins) and l is the maximum order of interactions considered. To do this, the authors introduce a q-ary extension of their previously-published binary Mobius transformation. The authors test their method on three existing sequence-to-function models (TIGER, inDelphi, and Transception). For TIGER and inDelphi, they show that their Fourier coefficients predict the test set better than linear or pairwise models and that their calculated Shapley values correlate well with those produced by KernelSHAP and SHAP-IQ while being orders of magnitude faster. For Transception, the authors additionally found epistatic interactions between certain residues of GFP.

**Questions:**

How does SHAP zero perform when asked to evaluate rare and highly deleterious mutations? Even if it doesn’t assign as highly-negative of values as a method like KernelSHAP, does it still correctly identify those mutations as being the most explanatory?

How does SHAP zero perform in the context of a sequence landscape that would be computationally-intractable for methods like KernelSHAP and SHAP-IQ? Suggestions include models trained on MPRA datasets or even a synthetic dataset where all the feature contributions and higher-order interactions are known. Even though it would be impossible to compare to slower methods, it would add to the paper’s significance if the method was able to pull out known feature contributions/interactions while also identifying new ones.

The Tranception case study is the least fleshed-out and most confusing, especially for epistatic interactions. The statement in the 4d figure legend “SHAP zero reveals numerous epistatic interactions with Proline (P) and Lysine (K).” is not actually shown in the figure (that I can tell) and needs to be inferred from Appendix D tables. It’s also unclear why the correlations with SHAP-IQ values are relatively low, and there is no analysis of whether the Fourier coefficients predict mutational effects better than a linear or pairwise baselines, so while SHAP zero is pulling out interactions, it’s hard to evaluate how trustworthy those conclusions are.

Model evaluation cost:  The initial sketch requires many queries for TIGER/inDelphi (appendix). For large transformer models whose forward pass dominates runtime, the wall‑clock break‑even point (e.g. 25 k sequences) may be restrictive for some users. Discussion of adaptive or incremental sketching could be beneficial.

**Ethical Concerns:**

["NO or VERY MINOR ethics concerns only"]

**Final Justification:**

I maintain my score having read the author's comments. Technically solid, high impact in one sub-area of AI, but not wide ranging enough to get a 6.

**Limitations:**

Yes, the authors adequately addressed the limitations of their work, except for some points that were addressed earlier.

**Quality:**

3

**Strengths And Weaknesses:**

Strengths:
- The paper was clear and well-motivated.
- The authors benchmarked their method in three different biological contexts, covering RNA, DNA, and protein sequences.
- Their q-ary Mobius transform allows for amortization in biological sequence contexts, filling in a “missing link”, so to speak.
- The paper compares to FastSHAP, which uses a neural network to amortise, and provides some insight into why FastSHAP does not work very well for biological sequence models.
- SHAP zero is impressively fast while maintaining decent-to-great correlation with slower methods, opening the door to interpreting sequences-to-function models that might have been intractable for slower methods.
- Theoretical analysis of the approach is provided, filling in details such as complexity.
Weaknesses:
- The sparse Fourier algorithm assumes equal probability of all feature subsets, which doesn’t reflect real sequence distributions. The authors recognize this as a limitation when they note that KernelSHAP has more negative values for rare token values, compared to SHAP zero, but they don’t fully flesh out the implications for applying their method since rare nucleotides/amino acids usually reflect strong negative selection. For example, it would be helpful for potential users to know how their method labels stop codons/CpGs/etc to aid in interpreting results.
- The major advantage of SHAP zero is the ability to assign Shapley values in contexts that were previously intractable, so it seems like a missed opportunity that the authors don’t apply their method to any such context since that’s where future users of the tool will likely want to use their method (the Transception case study is still in the context of a small GFP landscape).

---

> ### Author Rebuttal · Authors · 2025-07-29
>
> We thank reviewer hhqE for their comments. Below, we address the following questions:
>
> > How does SHAP zero perform when asked to evaluate rare and highly deleterious mutations? Even if it doesn’t assign as highly-negative of values as a method like KernelSHAP, does it still correctly identify those mutations as being the most explanatory?
>
> We clarify that Shapley estimation algorithms finds the most impactful mutations on a sequence, which includes both deleterious and beneficial mutations. While KernelSHAP tends to assign more negative values, the overall explanations SHAP zero returns is sufficient to identify rare and impactful mutations. SHAP zero’s explanations achieve a correlation of $\rho=0.97$ with KernelSHAP’s estimates, indicating that both methods agree on the relative importance of mutations. Examining the top mutations (Tables 8 and 9), SHAP zero identifies all the top positive mutations and 17 out of 20 of the top negative mutations found by KernelSHAP. The three negative interactions that were missed were all ranked in the bottom five of KernelSHAP's list. This demonstrates that while SHAP zero's negative value assignments are more conservative, it is still able to capture the most impactful mutations.
>
> We will add a summary of this to the Discussion section.
>
> > How does SHAP zero perform in the context of a sequence landscape that would be computationally-intractable for methods like KernelSHAP and SHAP-IQ? Suggestions include models trained on MPRA datasets or even a synthetic dataset where all the feature contributions and higher-order interactions are known. Even though it would be impossible to compare to slower methods, it would add to the paper’s significance if the method was able to pull out known feature contributions/interactions while also identifying new ones.
>
> SHAP zero will be able to recover meaningful interactions in landscapes that are intractable for methods like KernelSHAP and SHAP-IQ under the key assumption of Fourier sparsity. This means that only a small fraction of interactions are relevant to the model’s output. In the context of models trained on MPRA datasets, previous studies show that such models encode regulatory motifs which are known to be sparse in Fourier [1-2], suggesting that our method's core assumption holds in this domain.
>
> > it’s unclear why the correlations with SHAP-IQ values are relatively low… how trustworthy those conclusions [(SHAP zero on Tranception)] are… [with] linear or pairwise baselines.
>
> We clarify that the correlation between SHAP zero and SHAP-IQ is $\rho=0.61$, which demonstrates that the interactions recovered are in agreement with one another to some degree. Our analysis of the top biological motifs from Tables 8 and 10 shows that the top interactions recovered by SHAP zero corresponds to known motifs that affect protein stability. Regardless, to better assess the trustworthiness of SHAP zero, we have additionally performed two new experiments. First, we took the top 5 positive and negative interactions identified by SHAP zero from Table 10 and explicitly verified that these interactions are epistatic in Tranception. Second, we compared the performance of SHAP zero’s recovering Fourier coefficients against Tranception and linear and pairwise baselines (see Reviewer jhji for the full Table). We observe that SHAP zero ($\rho=0.55$) is able to closely approximate the performance of Tranception ($\rho=0.57$), suggesting that the epistatic interactions SHAP zero captures are biologically meaningful.
>
> We will add these new results to Section 4 and the Appendix and update Fig. 4 to better highlight the epistatic interactions discovered by SHAP zero.
>
> > Model evaluation cost… may be restrictive for some users
>
> The one-time cost required for SHAP zero's initial sketch is relatively low compared to the cost of training modern AI models. SHAP zero is designed to use less computational power in the long run by reducing the amortized cost of future explanations to near-zero.
>
> The model evaluation cost of SHAP zero is primarily determined by two factors: the query speed and the Fourier sparsity of the model. In the case of ML-guided protein design, it is common to screen for a pre-selected list of important sites [3-4], rather than over the entire sequence. When running SHAP zero for larger, slower-to-query models like Tranception, this helps keep the problem well-defined and computationally feasible. For analyses over a very large number of sites, an interesting extension would be to run multiple, smaller SHAP zero instances across different regions of the sequence, and combine the explanations. This approach would be more computationally efficient than a single, large run because each SHAP zero instance would only need to recover a smaller set of sparse Fourier coefficients.
>
> We will additionally release a pip package with the camera-ready version. This package will automatically tune SHAP zero’s parameters based on a user-defined sample budget (following our computational budget selection in Appendix E.1), providing the best possible Shapley estimates for their available resources.
>
> [1] Agarwal et al. “Massively parallel characterization of transcriptional regulatory elements” (2025)
>
> [2] Dalla-Torre et al. “Nucleotide Transformer: building and evaluating robust foundation models for human genomics” (2024)
>
> [3] Bian et al. “Optimizing enzyme thermostability by combining multiple mutations using protein language model” (2024)
>
> [4] Lipsh-Sokolik et al. “Addressing epistasis in the design of protein function” (2024)

---

### Official Review · Reviewer_jhji · 2025-07-02

**Clarity:** 4
**Significance:** 3
**Originality:** 3
**Rating:** 5
**Confidence:** 3

**Summary:**

This work develops a new approach for SHAP values estimation in black box biological sequence models, focusing on amortisation of computational cost. The approach is to, instead of querying SHAP values for every sequence of interest, build a global sparse representation of the model, which only needs to be done once, using a sparse Fourier transform, and connect this to sequence-specific SHAP values via a Möbius transformation. They test the method's ability to explain guide rNA binding (with TIGER), DNA repair outcome (with inDelphi), and epistasis in protein language models (with Tranception). They show good agreement with SOTA approaches, both at extracting SHAP values and high-order interactions, at a much lower computational cost, especially in the regime of small sequences and large number of queries. The article is well written and clear.

**Questions:**

1. If I understand correctly, you are querying variation on 10 positions known to be epistatic in GFP. The first question is how does your Fourier representation correlate with experiments, compared to Tranception, linear and pairwise representations (an equivalent to 2b and 3b)? Maybe I missed it.
I think that perhaps a more interesting question than looking at known interacting positions would be to, given many positions, identify the ones that are epistatically interacting. This would be more in line with tasks such as identifying allosteric sites involved in ligand binding, protein design, etc. Would this be computationally viable? (you would increase sequence length but would look at a sparser space) See question 2, related.
2. How far in sequence length can you take this approach? I think a scalability analysis in a protein and RNA/DNA setting would be useful.
3. Have you considered using other sparse representations such as wavelets instead of a sparse Fourier basis? What do you think would be the relative pros and cons?
4. If I understand correctly, in line 223 you say that the overlapping high-order feature interactions between SHAP zero and SHAP-IQ correspond to 95% of the total SHAP zero high-order feature interactions and 5% of those from SHAP-IQ. Why the difference in total number of high-order interactions between these two approaches? Also, why is it the opposite in the DNA repair example, line 244?

**Ethical Concerns:**

["NO or VERY MINOR ethics concerns only"]

**Final Justification:**

All issues were resolved. I recommend the authors to add a small discussion on scalability of the approach to the article, in line with the rebuttal. I am keeping the original acceptance score.

**Limitations:**

It would be useful to have a better picture of the scalability of the approach as mentioned in question 2.

**Paper Formatting Concerns:**

Did not notice any major issues.

**Quality:**

3

**Strengths And Weaknesses:**

Strengths:
Interesting idea, well exposed, with potential to be portable across multiple tasks with biological sequence models and have therefore broad adoption. The approach is technically solid and the experiments are for the most part sufficiently appropriate to be convincing of this strategy's value. I find it particularly interesting that the Fourier coefficients recovered from SHAP zero perform so well at the RNA binding and DNA repair tasks, strongly suggesting that this is a meaningful representation of the data.

Weaknesses:
The approach seems to have benefits in computational costs whenever the number of queries is large, the function sufficiently sparse, and the length of the input sequence is not too large. This questions how broadly adopted this strategy can be in practice.

I found that the protein language model experiment could have been taken further to be more representative of the kinds of tasks the community is interested in (see questions).

---

> ### Author Rebuttal · Authors · 2025-07-29
>
> We thank reviewer jhji for their constructive feedback. Below, we address specific questions:
>
> > how does your Fourier representation correlate with experiments compared to Tranception, linear and pairwise representations
>
> To address this question, we performed new experiments comparing SHAP zero against Tranception and linear/pairwise baselines on the deep mutational scanning data from [1], following the procedure in Appendix E.6:
> Model|Pearson|Spearman
> -|-|-
>  |Tranception|0.57|0.75
>  |SHAP zero|**0.55**|0.69
>  |Linear|0.48|0.54
>  |Pairwise|0.33|0.41
> SHAP zero outperforms the linear and pairwise baselines in generalizing to deep mutational scanning data. It also closely approximates the performance of the full Tranception model. We will add these results to our protein experiments in Section 4.
>
> > I think that perhaps a more interesting question than looking at known interacting positions would be to, given many positions, identify the ones that are epistatically interacting… Would this be computationally viable? (you would increase sequence length but would look at a sparser space)...  How far in sequence length can you take this approach?
>
> We agree that identifying epistatic positions at scale is an exciting future direction that SHAP zero is well-equipped to handle. The scalability of this approach depends both on the query speed and the relative sparsity of the model. On our single GPU setup, assuming a time budget of 10 days and that the relative sparsity of the model remains constant as we scale up $n$:
> - We can recover interactions across approximately $n=30$ sites for larger, slower-to-query models like Tranception.
> - For smaller, faster-to-query models like TIGER, the same budget would allow us to scale to $n=100$ sites.
>
> As $n$ grows, we speculate that the relative sparsity level will likely decrease, enabling SHAP zero to analyze significantly more positions than we report here.
>
> We additionally clarify that, in our current experimental setup, finding epistatic interactions given a subset of positions is an important and high impact problem [2-4], especially in the context of discovering proteins with novel biological motifs. Screening just 10 positions results in a combinatorial space of over 10 trillion sequences ($20^10 \sim 1 \times 10^13). Existing methods screen this landscape by restricting their search to small, local neighborhoods within the protein's 3D structure [2, 4] or over a small fraction of amino acids [3]. We emphasize that SHAP zero, unlike these methods, does not require such restrictive assumptions, which enables the recovery of epistatic interactions previously inaccessible at scale.
>
> > Have you considered using other sparse representations such as wavelets instead of a sparse Fourier basis?
>
> We have not considered other representations such as wavelets. In the context of biological sequence-function models, Fourier is the only standard basis which defines interactions in sequences, consistent with the definition of epistasis in statistical genetics [5]. SHAP zero could, in principle, be generalized to different bases, which would be an interesting future direction.
>
> > Why the difference in total number of high-order interactions between [SHAP zero and SHAP-IQ in the protein experiments]?
>
> Both SHAP zero and SHAP-IQ are approximating Faith-Shap interactions in Tranception. Given the computational budgets provided to both algorithms, we observe that SHAP zero identifies more biologically meaningful interactions. We attribute this success to SHAP zero’s underlying assumption of Fourier sparsity, a principle that previous studies suggest holds for biological fitness landscapes [6-8]. This assumption allows SHAP zero to structurally subsample the model to efficiently isolate key interactions. In contrast, SHAP-IQ's reliance on random feature subset sampling is less efficient for this problem and requires a larger computational budget to achieve comparable results.
>
> > Also, why is it the opposite in the DNA repair example, line 244?
>
> Thank you for pointing out the inconsistency in the DNA repair example—that is a typo on our part, and we apologize for the error. The trend is the same as in our protein experiments. The corrected text should read:
> “interactions (constituting 85% and 3% of the total interactions in **SHAP zero** and **SHAP-IQ**, respectively)”
>
> [1] Sarkisyan et al. "Local fitness landscape of the green fluorescent protein" (2016)
>
> [2] Weinstein et al. “Designed active-site library reveals thousands of functional GFP variants” (2023)
>
> [3] Wu et al. “Adaptation in protein fitness landscapes is facilitated by indirect paths” (2016)
>
> [4] Starr et al. “Epistasis in protein evolution” (2016)
>
> [5] Poelwijk et al. "The context-dependence of mutations: a linkage of formalisms" (2016)
>
> [6] Aghazadeh et al. “CRISPRLand: Interpretable large-scale inference of DNA repair landscape based on a spectral approach” (2020)
>
> [7] Brookes et al. "On the sparsity of fitness functions and implications for learning" (2022)
>
> [8] Tsui et al. “On recovering higher-order interactions from protein language models” (2024)

---

### Comment · Area_Chair_95hU · 2025-08-04

Dear reviewers, this paper needs discussion. Could you please engage asap?

---

### Note · Authors · 2025-08-12

We thank all the reviewers for their detailed and constructive feedback. We highlight a few key discussion points.

> Trustworthiness of SHAP zero in protein experiments (Tranception).

We addressed these comments by 1) running linear and pairwise baselines compared to SHAP zero’s Fourier representation, 2) examining the top mutations SHAP zero highlights and comparing them to KernelSHAP, and 3) explicitly verifying that the top SHAP zero interactions are epistatic in Tranception. These three experiments confirm that the SHAP values and interactions recovered by SHAP zero are biologically meaningful.

> Comparison of SHAP zero against SHAP-IQ.

Both SHAP zero and SHAP-IQ aim to approximate Faith-Shap interactions in biological sequence models. We observe that, given their respective computational budgets, SHAP zero and SHAP-IQ interactions, to some extent, agree. However, SHAP zero identifies more biologically meaningful interactions, as shown in our results and additional experiments. SHAP zero, due to its underlying assumption of Fourier sparsity, is able to isolate key interactions within biological sequence models.

> Applicability of SHAP zero in other biological problems.

We appreciate the reviewers’ interest in other applications of SHAP zero. We agree that there are many exciting future directions that SHAP zero is well-equipped to handle. In other combinatorial biological problems such as the identification of allosteric sites, explaining models trained on MPRA datasets, and uncovering protein-protein interactions, SHAP zero can be seamlessly applied.

We additionally clarify that our current experiments tackle high impact problems in DNA, RNA, and protein sequence models. Our results demonstrate that SHAP zero is capable of identifying biological motifs at scale, which was previously inaccessible.

Overall, we believe we have addressed all the reviewers’ concerns and questions to the best of our ability. We greatly appreciate the thorough reviews, and we will use the feedback given to update the camera-ready paper appropriately.

---

### Decision · Program_Chairs · 2025-09-17

**Decision:**

Accept (poster)

**Comment:**

This paper introduces SHAP zero, an algorithm for explaining black-box biological sequence models. By amortizing the computational cost of calculating Shapley values across many queries, the method enables large-scale interpretability studies that were previously intractable. The main idea is to create a one-time global sketch of the model using a sparse Fourier transform, from which local explanations for future queries can be derived at near-zero marginal cost. The authors demonstrate its effectiveness on models for gRNA efficacy, DNA repair, and protein fitness.

The reviewers agreed that the paper is very well written, and that the methods presented are exciting and important, and also the core idea is neat and clever. The discussion clarified reviewers questions, and there the paper seems to be at the level suitable for this conference.